# MicroRNA-mediated repression of nonsense mRNAs

Ya Zhao[1,2], Jimin Lin[1,2], Beiying Xu[1,2], Sida Hu[1,2], Xue Zhang[1,2], Ligang Wu[1,2]*

[1]State Key Laboratory of Molecular Biology, Institute of Biochemistry and Cell Biology, Shanghai Institutes for Biological Sciences, Chinese Academy of Sciences, Shanghai, China; [2]Shanghai Key Laboratory of Molecular Andrology, Institute of Biochemistry and Cell Biology, Shanghai Institutes for Biological Sciences, Chinese Academy of Sciences, Shanghai, China

**Abstract** Numerous studies have established important roles for microRNAs (miRNAs) in regulating gene expression. Here, we report that miRNAs also serve as a surveillance system to repress the expression of nonsense mRNAs that may produce harmful truncated proteins. Upon recognition of the premature termination codon by the translating ribosome, the downstream portion of the coding region of an mRNA is redefined as part of the 3′ untranslated region; as a result, the miRNA-responsive elements embedded in this region can be detected by miRNAs, triggering accelerated mRNA deadenylation and translational inhibition. We demonstrate that naturally occurring cancer-causing *APC* (adenomatous polyposis coli) nonsense mutants which escape nonsense-mediated mRNA decay (NMD) are repressed by miRNA-mediated surveillance. In addition, we show that miRNA-mediated surveillance and exon–exon junction complex-mediated NMD are not mutually exclusive and act additively to enhance the repressive activity. Therefore, we have uncovered a new role for miRNAs in repressing nonsense mutant mRNAs.

*For correspondence: lgwu@sibcb.ac.cn

**Competing interests:** The authors declare that no competing interests exist.

**Reviewing editor**: Nick J Proudfoot, University of Oxford, United Kingdom

## Introduction

Eukaryotic cells are constantly at risk for various types of mutations. Although many of the mutations are benign, a high number of mutations have detrimental consequences. Among these mutations, the nonsense mutation is a severe type that converts a coding codon into a stop codon, leading to the premature termination of translation and the expression of proteins truncated at the carboxyl terminus. These truncated protein products often have deleterious dominant-negative or gain-of-function effects that interfere with normal biological processes in cells. Indeed, many inherited genetic disorders, such as β-thalassemia (*Chang and Kan, 1979*) and Duchenne muscular dystrophy (*Koenig et al., 1987*; *Monaco et al., 1988*), are caused by germline nonsense mutations. Moreover, nonsense mutations in critical tumor suppressor genes are associated with prevalent cancer types such as breast cancer (*Miki et al., 1994*) and colorectal cancer (*Powell et al., 1992*; *Rowan et al., 2000*). A recent large-scale genome-wide study revealed that even healthy individuals carry dozens of nonsense mutations (*MacArthur et al., 2012*). In addition, transcriptional errors, mis-splicing, or even alternative splicing (*Danckwardt et al., 2002*; *Lewis et al., 2003*; *Wollerton et al., 2004*) also frequently lead to nonsense mutations. Accordingly, cells have evolved a surveillance system known as nonsense-mediated mRNA decay (NMD) to eliminate these aberrant transcripts.

Great efforts have been made to uncover the mechanism by which cells detect and selectively degrade nonsense mRNAs. One well known mechanism for recognizing premature termination codon (PTC)-containing transcripts is exon–exon junction complex (EJC)-dependent NMD (EJC-NMD). During splicing, a multi-protein EJC complex is deposited at the 5′ side of each exon–exon junction, which is subsequently displaced by the translating ribosome during the pioneer round of translation.

**eLife digest** To produce a protein from a gene, the sequence of the gene must be transcribed to produce a molecule of messenger RNA (mRNA). The sequence of the mRNA is then read in groups of three letters at a time (called codons), and each codon instructs for a particular amino acid to be added into the protein. Some codons, however, do not code for an amino acid and instead these 'stop codons' mark the end of a protein.

If a DNA letter is added, lost, or changed, this mutation can sometimes produce a stop codon too early in the mRNA sequence. This is called a nonsense mutation, and produces truncated proteins that either work incorrectly or do not work at all, which can harm the organism. For example, people with a nonsense mutation in the human tumor suppressor gene called *APC*—which normally stops uncontrolled cell growth and division—are more likely to develop colon cancer than people without this mutation.

Cells in the body employ several different surveillance mechanisms to detect nonsense mutations. The best-known mechanism involves a large protein group called the exon–exon junction complex (EJC), which binds to sites within the mRNA. The cellular translation machinery removes all the EJCs bound to a normal mRNA during the production of proteins. If the translation machinery reaches a stop codon too early, so that EJCs located downstream of it are not removed, the mRNA molecule is destroyed. However, this mechanism does not work for all genes—including *APC*.

Very short sections of RNA called microRNAs regulate protein production by causing mRNAs to degrade and by inhibiting their translation, and Zhao et al. have now found that microRNAs also act as a defense against nonsense mutations in the *APC* gene. A premature stop codon exposes sites further along the mRNA molecule that microRNA molecules bind to, which triggers the breaking down of the mRNA and inhibits its translation. The microRNA surveillance system works independently of the system involving the EJC. However, both mechanisms can work in parallel alongside each other, which provides extra protection against nonsense mutations.

Zhao et al. also found that microRNAs can protect against nonsense mutations in several other types of gene found in human cells. Therefore, microRNA surveillance is likely to be a common method employed by cells to restrict the production of potentially harmful truncated proteins.

An EJC will remain bound to the mRNA and trigger NMD efficiently if the ribosome stalls at a PTC located at least 50 nucleotides (nt) upstream of the final exon–exon junction (*Maquat, 2004*). However, nonsense transcripts that originate from naturally intronless genes are immune to EJC-NMD, as are transcripts with PTCs located within the last exon or less than 50 nt upstream of the last exon–exon junction. Other EJC-independent NMD mechanisms, such as long 3' untranslated regions (UTRs) (*Buhler et al., 2006*; *Eberle et al., 2008*; *Singh et al., 2008*; *Hogg and Goff, 2010*; *Zund et al., 2013*) and upstream open reading frames (uORFs) (*Matsui et al., 2007*) are also involved in the degradation of PTC-containing mRNAs. The diverse degradation pathways for nonsense mRNAs indicate the complexity of mRNA quality control mechanisms in living cells.

Another mechanism for post-transcriptional gene regulation is mediated by microRNAs (miRNAs), a class of small non-coding RNAs that are present in essentially every organ and tissue of the body. In animals, miRNAs are processed from hairpin precursors and assemble with Argonaute (Ago) family proteins into RNA-induced silencing complexes (RISCs) to regulate gene expression (*Bartel, 2004*). By imperfectly base-pairing with miRNA-responsive elements (miREs) in target mRNAs (primarily through nucleotides 2–7, the seed region of the miRNA), miRNAs exert their repressive effects by promoting RNA degradation through accelerated deadenylation and/or by inhibiting translation (*Wu and Belasco, 2008b*). The richness and variety of cellular miRNAs, as well as the versatility of miRNA:miRE seed matches render miRNAs one of the most flexible molecules to govern transcriptome integrity.

In this study, we demonstrate that miRNAs also selectively target and repress the expression of nonsense mRNAs by both expedited poly(A) tail removal and translational repression. We present evidence that naturally occurring cancer-causing nonsense mRNAs are repressed by miRNA-mediated surveillance. Furthermore, we show that miRNA-mediated surveillance and EJC-NMD function additively. We propose that miRNAs may serve as a novel component of the cellular mRNA quality control system that eliminates nonsense mRNAs.

# Results

## A PTC potentiates miRNA-mediated repression of nonsense mRNAs

The vast majority of known functional and conserved miREs reside within the 3′ UTR of mRNAs (*Bartel, 2004*). By contrast, reports of miRNAs efficiently targeting ORFs are sparse (*Duursma et al., 2008*; *Forman et al., 2008*; *Tay et al., 2008*; *Huang et al., 2010*; *Schnall-Levin et al., 2011*). Interestingly, recent studies describing the transcriptome-wide identification of miREs revealed prevalent RISC binding in the coding region (*Chi et al., 2009*; *Hafner et al., 2010*; *Helwak et al., 2013*), raising a fascinating question about the biological significance of these ORF miREs. Previously, we have shown that miRNAs cause expedited removal of the poly(A) tails from their mRNA targets through the recognition of miREs in the 3′ UTR. By accelerating this initial and rate-limiting step of mRNA decay, miRNAs efficiently reduce the cellular concentration of their target mRNAs (*Wu et al., 2006*; *Figure 1—figure supplement 2A*). Theoretically, premature translation termination at a nonsense mutation should cause the ORF region downstream of the PTC to acquire a 3′ UTR identity. If any functional miRE is located within this redefined 3′ UTR region, this nonsense mRNA may become miRNA-sensitive and be subject to rapid miRNA-mediated deadenylation and then decay. In this manner, miRNAs could serve as a surveillance system against nonsense mRNAs.

To test this hypothesis, we used a transiently inducible β-globin (BG) reporter system and a well-established transcriptional pulse-chase assay to analyze the effect of a PTC on mRNA deadenylation in human cells (*Shyu et al., 1989*). Briefly, transcription was induced by removing tetracycline (tet) from the culture medium for a short period so as to obtain a nearly homogeneous population of BG mRNAs that subsequently underwent synchronous deadenylation and degradation. RNA samples collected at different time points after induction were subjected to site-specific cleavage by RNase H to produce 3′ BG mRNA fragments which facilitate accurate measurement of the poly(A) tail length via gel electrophoresis and Northern blotting. The natural sequence of BG mRNA does not harbor any miREs; therefore, we first modified the reporter by inserting a let-7a miRE sequence (*Figure 1F*) in-frame into the ORF of the last exon to create LastEx-L7. We found that this insertion did not significantly accelerate poly(A) shortening in HeLa-tTA cells, where let-7a is naturally abundant (*Figure 1A*, compare LastEx-L7 and TBG). This result indicates that ORF miREs are not functionally efficient. Interestingly, a point mutation (AAA to TAA) in the last exon that introduced a PTC 16 nt upstream of the let-7a miRE of LastEx-L7 (LastEx-PTC-L7) significantly accelerated the shortening of the 3′ fragment (*Figure 1A*, upper portion, compare LastEx-L7 and LastEx-PTC-L7, time points 3 and 4.5), but not the 5′ fragment, of BG mRNA (*Figure 1A*, bottom portion), a finding indicative of expedited poly(A) removal. Furthermore, treatment of the RNA samples with oligo(dT) and RNase H caused the BG 3′ fragments that previously appeared as diffuse bands after electrophoresis to migrate uniformly to a position corresponding to fully deadenylated mRNA (*Figure 1B*), which constitutes additional evidence that the decrease in the length of BG 3′ fragments was due to trimming of the poly(A) tails. The expedited deadenylation disappeared when the let-7a miRE was mutated (LastEx-PTC-L7M), indicating that the rapid poly(A) removal was specifically induced by endogenous let-7a in the cells (*Figure 1A*). As a consequence of accelerated deadenylation, the BG mRNA bearing both the let-7a miRE and the upstream PTC decayed much faster than counterparts that lacked the PTC or the let-7a miRE or bore the mutant let-7a miRE, with the half-life decreasing from >6 hr to <3 hr (*Figure 1—figure supplement 1A,B*). LastEx-L7 decayed slightly faster than the control BG mRNAs (TBG and LastEx-PTC), which suggests that the miRE located in the ORF may still have residual activity. Similarly, the deadenylation rate of another BG reporter with a PTC located 11 nt upstream of the last exon–exon junction was also accelerated in a miRNA-dependent manner (*Figure 1C*, compare PTC102-L7 and PTC102-L7M, time points 1.5, 3, and 4.5). Expedited poly(A) shortening was not observed for BG mRNA bearing the PTC in the last exon or 11 nt upstream of the last exon–exon junction but no downstream miRE (*Figure 1A*, LastEx-PTC and *Figure 1C*, PTC102), because the PTC alone is unable to trigger EJC-NMD when located in the last exon or less than 50 nt upstream of the last exon–exon junction. Similar results were obtained with BG reporters containing one miR-21 miRE (*Figure 1—figure supplement 2B,C* and data not shown), providing evidence that miRNA-mediated deadenylation of nonsense mRNAs is not miRNA-type-specific.

Besides point mutations, alternative splicing also serves as an important source of PTCs. Recent studies suggest that intron retention, a splicing event that often creates PTCs, functions as a general mechanism that controls the expression of many genes in the cells (*Galante et al., 2004*; *Yap et al.,*

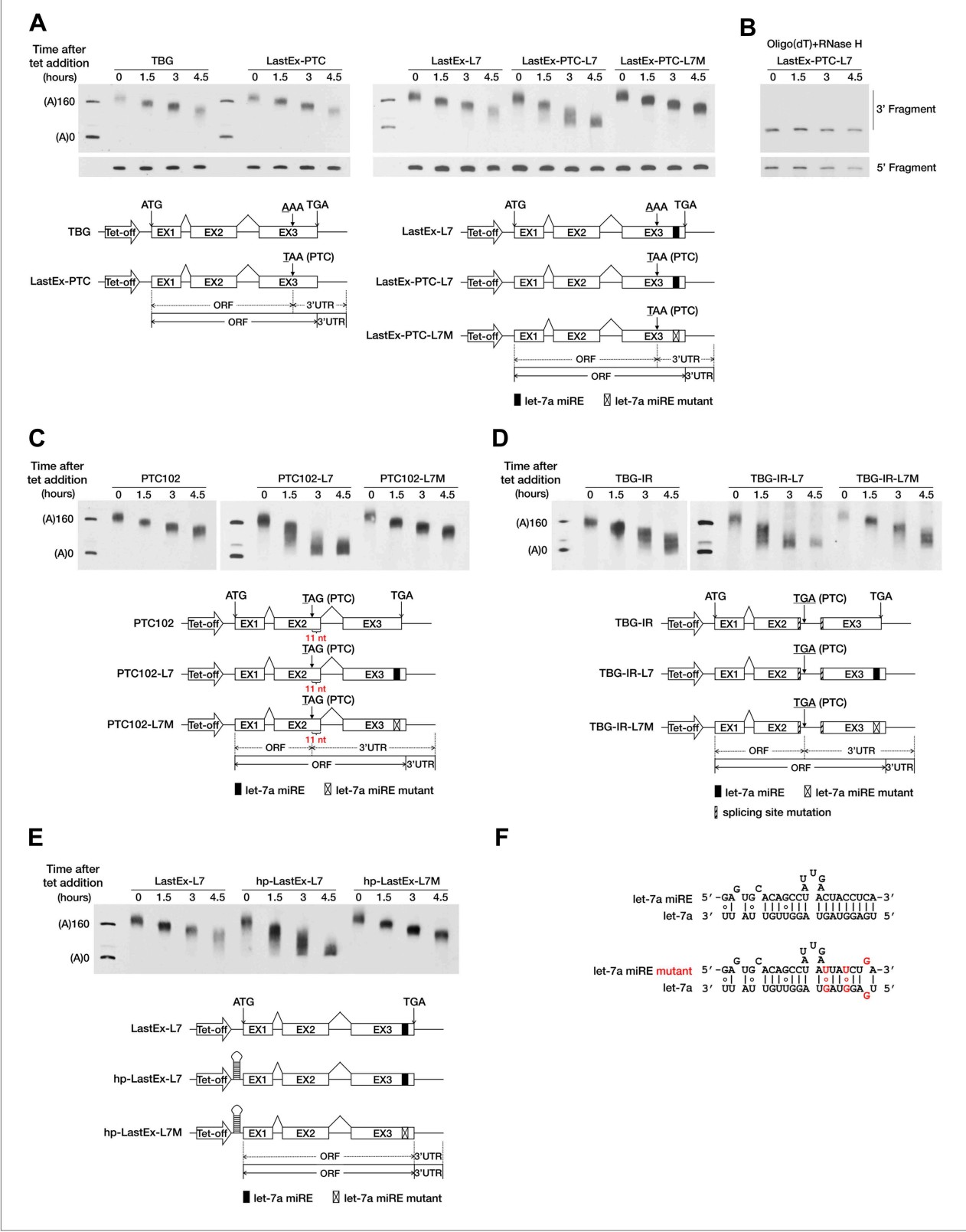

**Figure 1**. A PTC potentiates rapid miRNA-mediated deadenylation of nonsense mRNAs. (**A**) The influence of a PTC on BG mRNAs with or without a downstream miRE. Cytoplasmic RNA was collected at the indicated times after transcriptional arrest by adding tetracycline (tet). The RNA samples were then treated with RNase H and an oligodeoxynucleotide complementary to codons 74–81 of BG mRNA to generate 5' and 3' fragments, which were then

*Figure 1. Continued on next page*

Figure 1. Continued

separated by electrophoresis on a denaturing polyacrylamide gel and detected by Northern blotting. Left panel: BG mRNA deadenylation with or without a PTC in the last exon. TBG contains an intact BG ORF. A PTC was introduced into TBG at codon 121 within the last exon to generate LastEx-PTC. Both constructs harbor no miRE in their ORFs. Right panel: the influence of let-7a on the deadenylation rate of BG mRNAs harboring one let-7a miRE in its ORF with or without an upstream PTC mutation. LastEx-L7 contains one let-7a miRE in the last exon of the BG ORF. LastEx-PTC-L7 has a PTC at the same position as LastEx-PTC, which is 16 nt upstream of the let-7a miRE. Two nucleotides within LastEx-PTC-L7 let-7a miRE seed were changed to create a synonymous codon in LastEx-PTC-L7M. The positions of the ORF start site and original stop codon are indicated by arrows. The borders of the original ORF/3′ UTR and redefined ORF/3′ UTR upon PTC mutation are indicated by solid or dashed lines below the constructs. Markers A(0) and A(160) correspond in size to BG mRNA 3′ fragments bearing no poly(A) or a 160-nt poly(A) tail, respectively. (**B**) Confirmation of poly(A) tail shortening by treatment with oligo(dT) and RNase H. The same LastEx-PTC-L7 RNA as in **A** were further treated with oligo(dT) and RNase H and analyzed by Northern blotting. (**C**) Induction by let-7a of accelerated deadenylation of nonsense BG mRNAs that do not conform to the '50 nt boundary rule' of EJC-NMD. PTC102-L7 contains a PTC mutation 11 nt upstream of the last exon–exon junction and a let-7a miRE in the last exon of the ORF. PTC102-L7M is identical to PTC102-L7 except for a mutated let-7a miRE. PTC102 contains the PTC only. (**D**) The influence of let-7a on the deadenylation rate of BG mRNAs harboring one let-7a miRE in its ORF with a retained last intron. TBG-IR-L7 contains one let-7a miRE in the last exon of the BG ORF and a retained last intron that creates a PTC 505 nt upstream of the let-7a miRE. Two nucleotides within the TBG-IR-L7 let-7a miRE seed were mutated as in **A**. TBG-IR only has the retained intron but no miRE. (**E**) The influence of let-7a on the deadenylation rate of BG mRNAs harboring one let-7a miRE in its ORF in the absence of translation. hp-LastEx-L7 contains a 40-nt inverted repeat in its 5′ UTR to block translation initiation. hp-LastEx-L7M is identical to hp-LastEx-L7 except for a mutated let-7a miRE. (**F**) Duplexes expected for the let-7a miRE or its mutant counterpart base-paired with let-7a.

The following figure supplements are available for figure 1:

**Figure supplement 1**. A PTC potentiates rapid miRNA-mediated decay of nonsense mRNAs.

**Figure supplement 2**. A PTC potentiates rapid miR-21-mediated deadenylation of nonsense mRNAs.

**Figure supplement 3**. A large stem-loop structure in the 5′ UTR blocks translation of BG mRNA.

*2012*; *Wong et al., 2013*). We speculated that the PTC-containing transcripts generated by intron retention could also be targeted by miRNAs. To test this possibility, we generated a BG reporter (TBG-IR-L7) by mutating the splicing sites in the last intron of LastEx-L7, which abolished the splicing of the last intron and created a PTC 505 nt upstream of the let-7a miRE. As expected, we found that TBG-IR-L7 underwent rapid deadenylation in a miRNA-dependent manner; and that absence of the downstream miRE (TBG-IR) or mutations in the miRE seed region (TBG-IR-L7M) completely abolished this accelerated deadenylation (*Figure 1D*). Interestingly, TBG-IR is deadenylated slightly faster than wild-type TBG, probably due to the presence of potential miREs in the longer region between the PTC and the native stop codon (570 nt) that resulted from intron retention; these miREs may be recognized by the highly expressed endogenous miRNAs in HeLa-tTA cells, such as miR-10a/b and miR-17/20a (data not shown). Collectively, these observations demonstrate that a PTC, introduced either by point mutation or alternative splicing, is able to potentiate miRNA-mediated rapid deadenylation of the mRNA by unmasking ORF miREs downstream of it.

An essential feature that distinguishes the 3′ UTR from the ORF is the absence of translating ribosomes. The PTC may serve as a roadblock to stop ribosomes and mark the new boundary between the ORF and 3′ UTR. We speculated that blocking translation would cause the miRE within the ORF to behave as if it were in the 3′ UTR and efficiently trigger accelerated miRNA-mediated mRNA deadenylation, even in the absence of an upstream PTC. To test this hypothesis, we placed a large stem-loop structure at the 5′ UTR of LastEx-L7 (hp-LastEx-L7) to block translation initiation (*Chen et al., 1995*; *Wu et al., 2006*; *Figure 1—figure supplement 3*) and examined its deadenylation rate. As expected, blocking translation in this manner caused LastEx-L7 mRNA, which normally is deadenylated and decays slowly, to undergo rapid miRNA-mediated deadenylation and decay (half-life of >6 hr vs <3 hr) (*Figure 1E*, *Figure 1—figure supplement 1A,B*), suggesting that translating ribosomes may indeed have interfered with miRNA-RISC binding to the miREs located in the ORF and thus masked their repressive function. Similar results were obtained with another set of BG reporters containing a miR-21 miRE (*Figure 1—figure supplement 2D*). Together, these data suggest that miRNAs selectively accelerate the deadenylation of nonsense mRNAs by stably binding to miREs in the ORF downstream of the PTC. The immunity of PTC-free mRNA to miRNA-mediated repression may be due to masking of ORF miREs by the translating ribosomes, which is consistent with a previous study using constitutively transcribed luciferase reporters (*Gu et al., 2009*).

NMD exerts its repressive power primarily by promoting RNA degradation; however, miRNA-mediated repression usually involves both mRNA decay and translational repression. To determine whether translational repression is involved when miRNAs exert their repressive effects against nonsense mRNAs, we constructed a modified luciferase reporter that has a fragment containing two in-frame miR-125b miREs (*Figure 2A*) followed by an additional in-frame stop codon fused to the 3' end of the luciferase ORF (TAA-2E). In this construct, the original stop codon of the luciferase ORF served as a PTC. This PTC (TAA) was mutated to TCA to create a PTC-free counterpart (TCA-2E) (*Figure 2B*), which produced a luciferase protein with an additional 46 amino acids at the carboxyl terminal. Measurement of the steady-state mRNA level by qRT-PCR showed specific and significant repression of TAA-2E in the presence of miR-125b (*Figure 2C*), observations consistent with the results of the BG deadenylation assay shown in *Figure 1A* and *Figure 1—figure supplement 2B*. Moreover, the measurement of luciferase activity revealed an even greater reduction of the PTC-containing reporter at the protein level (*Figure 2D*), indicating that translational repression plays a prominent role in the repression of nonsense messages by miRNAs (*Figure 2E*). Since no intron is present downstream of the PTC, this reporter (TAA-2E) is immune to EJC-NMD and the repression is most likely contributed by miRNAs.

EJC-NMD generally complies with the '50 nt boundary rule'. To determine whether any boundary rule for miRNA-mediated surveillance exists, we constructed a series of plasmids in which a miR-125b miRE was inserted in-frame at various locations before or after the PTC of a modified luciferase reporter mRNA (*Figure 2F*). Measurement of luciferase activity revealed that an miRE has to be located at least 10 nt downstream of the PTC to trigger miRNA-mediated repression effectively (*Figure 2G*). This observation defines a distinct boundary rule for miRNAs to successfully repress PTC-containing messages and also suggests that the size of the footprint of RISC is much smaller than that of the EJC, which makes miRNA-mediated surveillance more versatile in recognizing and repressing nonsense mRNAs. Altogether, these results demonstrate that a PTC can potentiate miRNA-mediated deadenylation and translational inhibition of nonsense mRNAs, via redefinition of ORF/3' UTR identities and unmasking of downstream miREs.

## PTC-containing *APC* mRNAs are natural substrates repressed by miRNA-mediated surveillance

Next, we asked whether any naturally occurring nonsense mRNAs are subjected to regulation by miRNA-mediated surveillance. We found *APC* (adenomatous polyposis coli), a tumor suppressor gene that is frequently mutated in colorectal cancer, to be of particular interest. Most of the mutations in the *APC* gene that have been identified in clinical studies are point mutations or frameshift indels that create a PTC and result in the expression of a truncated version of the APC protein. Interestingly, the majority of known *APC* mutations are clustered in a hotspot region (designated as the MCR in *Figure 3A*) within the last exon of the ORF (*Miyoshi et al., 1992*), rendering these mutants immune to EJC-NMD. Meanwhile, bioinformatic analysis based on the seed match rule predicts numerous potential miREs between the MCR and the native stop codon in *APC* mRNA. All of these features make *APC* a nearly ideal paradigm for the study of miRNA-mediated surveillance.

To investigate whether PTC-containing *APC* mRNA is specifically targeted by certain miRNAs, we performed miRE screening using a reporter that has the region between a PTC at codon 1450 and the native stop codon (the PTC-STOP region) of *APC* mRNA fused to the 3' end of the luciferase ORF (*Figure 3—figure supplement 1A*). The top 45 miRNA candidates were chosen for the screening based on their general abundance in human tissues and the predicted thermal stability of the duplex they may form with an *APC* miRE. For each miRNA selected, we co-transfected HEK293 cells with the luciferase reporter plasmid and a synthetic miRNA mimic or a control small RNA, and examined protein production by measuring luciferase activity 36 hr after transfection. A decrease in luciferase activity when co-transfected with a miRNA mimic would indicate that the selected miRNA may have the potential to repress PTC-containing *APC* (PTC-*APC*) expression through miRE(s) in the PTC-STOP region. Using this method, we identified several miRNAs with a strong repressive effect (*Supplementary file 1*), although others that are naturally highly expressed in HEK293 cells even without transfection may have been missed. To verify that the repression is due to the direct interaction between the selected miRNA and an miRE in the PTC-*APC* mRNA, we mapped the corresponding miREs of the miRNAs via the 2–7 seed match alignment (*Figure 3A,I*, *Figure 3—figure supplement 1B*), and then inserted the ~30-nt-long sequence surrounding the predicted miREs into the 3' UTR of a luciferase

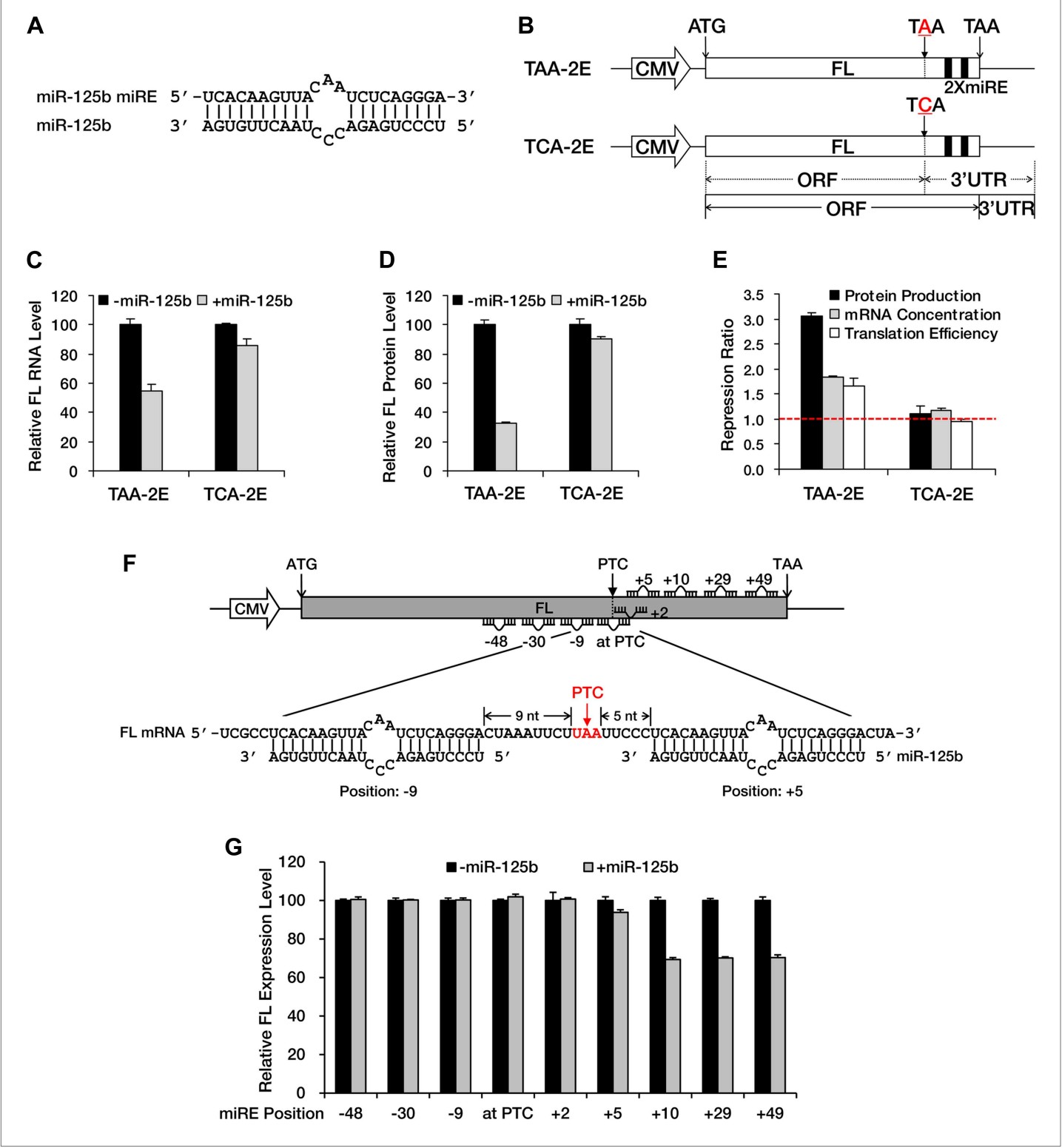

**Figure 2**. A PTC potentiates miRNA-mediated translational repression of nonsense mRNAs. (**A**) Duplex expected for the miR-125b miRE base-paired with miR-125b. (**B**) The reporter constructs used in **C** to **E**. TAA-2E contains two miR-125b miREs 69 nt downstream of the original stop codon (which now serves as a PTC) of the firefly luciferase (FL) gene. This PTC (TAA) was mutated to codon TCA to generate TCA-2E. The borders of the ORF/3′ UTR and redefined ORF/3′ UTR upon PTC mutation are indicated by solid or dashed lines below the constructs. (**C**) mRNA quantification for TAA-2E and TCA-2E in the presence and absence of miR-125b, as determined by qRT-PCR. The error bars represent the standard deviation of multiple measurements. (**D**)
*Figure 2. Continued on next page*

*Figure 2. Continued*

Protein quantification by analyzing luciferase activity for TAA-2E and TCA-2E in the presence and absence of miR-125b. (**E**) Contribution of translational repression by miRNA-mediated surveillance. The repression ratios for TAA-2E and TCA-2E were calculated from normalized levels of firefly luciferase protein (black bars) and mRNA (gray bars) in the absence versus the presence of miR-125b. By dividing the repression ratio for protein production and that for mRNA concentration, the repression ratio for translation efficiency (protein yield per mRNA molecule, white bars) was determined. A repression ratio for translation efficiency that is >1 indicates that part of the total repression observed at the protein level is attributable to inhibition of translation. (**F**) Schematic illustration of the reporter designs used in **G**. The same 22-nt miR-125b miRE as in **A** was introduced in-frame into different positions before or after a PTC of a modified firefly luciferase reporter gene to obtain a series of miRE-containing plasmids. A graph that illustrates the different methods for calculating the distance between an upstream or a downstream miRE and the PTC is shown below the construct. (**G**) Boundary rule for miRNA-mediated surveillance. Each firefly luciferase construct that contains one miR-125b miRE at a different position was co-transfected with a Renilla luciferase reporter into HEK293 cells in the presence or absence of miR-125b. The relative FL expression level was calculated from the normalized levels of firefly luciferase in the absence versus the presence of miR-125b.

reporter. Multiple miREs exhibited specific responses to the cognate miRNAs (*Figure 3B*), which verifies their functionality. To further confirm the repressive effect of these miRNAs on PTC-*APC* mRNA in its natural sequence context, we constructed minigene vectors that express both an HA-tagged PTC-*APC* bearing either a wild-type miRE or a mutant miRE with mismatches in the seed region and an HA-tagged *EGFP* that served as an internal control for more precise protein quantification. The minigene plasmids were co-transfected with cognate miRNA mimics into HEK293 cells. Western blotting revealed a marked increase in protein expression for the minigene constructs with mutant miREs (*Figure 3C*).

The unmasking of ORF miREs by PTCs may significantly augment repression by miREs in the 3′ UTR. To test this hypothesis, we constructed a pair of PTC-containing *APC* minigene plasmids that each contained the full-length *APC* 3′ UTR, one with wild-type miR-29a miREs (APC-PTC1450-3′UTR) and the other with mutant miREs (APC-PTC1450-mut-3′UTR). Western blotting showed that, in the context of the natural *APC* 3′ UTR, the PTC was still able to potentiate repression by miR-29a miREs originally located in the ORF (*Figure 3—figure supplement 2*). In addition, to quantify the relationship between unmasked ORF miREs and pre-existing miREs in the 3′ UTR, we designed chimeric reporters in which the PTC-STOP region and the entire 3′ UTR of *APC* mRNA were fused to the 3′ end of a firefly luciferase ORF. The miR-29a miREs in the PTC-STOP region and a miR-135b miRE in the 3′ UTR of *APC* mRNA that had previously been reported to be functional (*Nagel et al., 2008*) were mutated, either individually or simultaneously. In the presence of both miRNAs, the wild-type chimeric reporter was repressed most efficiently, while mutating either the ORF miREs or the 3′ UTR miRE alleviated the repression (*Figure 3—figure supplement 3*). These observations indicate that ORF miREs unmasked by an upstream PTC are fully functional in the presence of repression by 3′ UTR miREs.

We next sought to determine whether the expression of an endogenous PTC-*APC* mutant is downregulated by miRNAs. Our previous luciferase reporter- and minigene-based assays have identified that two miR-29a miREs (*Figure 3I*) are present in the PTC-STOP region of *APC* nonsense mRNA (*Figure 3A–C*). Interestingly, the seed regions of the miR-29a miREs embedded in the *APC* ORF are highly conserved across several species. Therefore, miREs of miR-29a were selected for subsequent investigations. SW480 is a colorectal cell line that naturally expresses a truncated APC protein (caused by a PTC mutation at codon 1338) that can be readily detected by Western blotting with a specific antibody (*Figure 3D*), and high levels of endogenous miR-29a (*Figure 3F,G*, *Figure 3—figure supplement 4B,D*), which render it a suitable cell line to investigate the repression mediated by miR-29a on the endogenous *APC* nonsense mutant. We transduced SW480 cells with lentiviruses encoding a TuD miRNA decoy (*Figure 3—figure supplement 4A*), which has been proven a very effective and specific antagonizer of miRNAs (*Haraguchi et al., 2009*), to inhibit endogenous miR-29a. TuD-expressing cells were cultured for 3 days before APC was examined by Western blotting. An approximate twofold increase in truncated APC expression was observed for SW480 cells in which miR-29a was knocked down (*Figure 3F*, upper panel). We also measured the mRNA level of PTC-*APC* in SW480 cells and found that the knockdown of miR-29a caused a ~1.6-fold increase of PTC-*APC* mRNA (*Figure 3E*), which is consistent with the important role of translational repression by miRNAs. The successful knockdown of endogenous miR-29a expression was confirmed by Northern blotting (*Figure 3F*, middle panel) and a reporter assay (*Figure 3—figure supplement 4B,C*); by contrast, the abundance of an untargeted endogenous miRNA, miR-26a, remained unchanged (*Figure 3F*, bottom

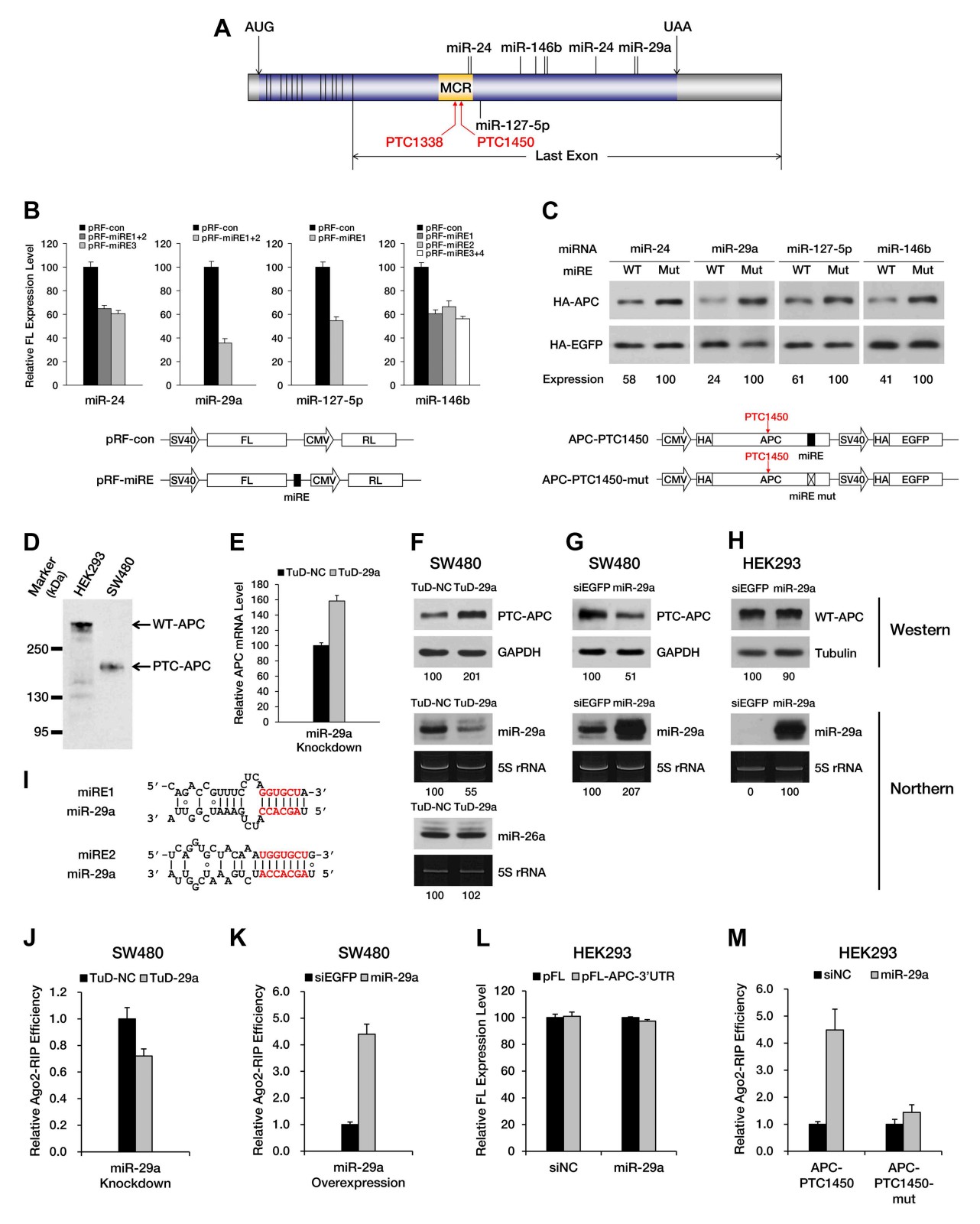

**Figure 3**. PTC-containing *APC* mRNAs are natural substrates repressed by miRNA-mediated surveillance. (**A**) Schematic representation of *APC* mRNA. The ORF is shown in blue. The 5′ UTR and 3′ UTR are shown in gray. Exon–exon boundaries are indicated by vertical lines in the mRNA. The positions of representative miREs and PTC sites are indicated above or below the mRNA by solid lines and arrows, respectively. MCR refers to the mutation cluster

*Figure 3. Continued on next page*

*Figure 3. Continued*

region. (**B**) Validation of miRE function by luciferase assays. A vector expressing both a firefly luciferase (FL) transcript harboring one potential miRE in its 3′ UTR and a control Renilla luciferase (RL) transcript was co-transfected with cognate miRNA mimics into HEK293 cells. The relative FL expression level represents the firefly/Renilla luciferase ratio for pRF-miRE relative to the no miRE control pRF-con. (**C**) Validation of miRE function in an *APC* minigene. A vector expressing both an HA-tagged truncated APC (APC-PTC1450) and a control HA-tagged EGFP was co-transfected with cognate miRNA mimics into HEK293 cells. The mutant counterpart of the *APC* minigene (APC-PTC1450-mut) contains two altered nucleotides that abolish miRE:miRNA complementarity without changing the identity of the encoded amino acid. (**D**) Western blot analysis of endogenous truncated APC in SW480 cells and full-length APC in HEK293 cells by using an anti-APC antibody. (**E**) Change in PTC-*APC* mRNA levels in SW480 cells upon miR-29a knockdown. Cytoplasmic RNA was extracted from the SW480 cell lines used in **F**, and the levels of *APC* and *GAPDH* mRNA were determined by qRT-PCR. The relative *APC* mRNA level was calculated by normalizing to *GAPDH* mRNA. (**F**) Upregulation of endogenous truncated APC upon miR-29a knockdown. SW480 cells were transduced with lentiviruses encoding a miR-29a decoy (TuD-29a) or a control decoy (TuD-NC). Endogenous truncated APC was probed with an anti-APC antibody. GAPDH served as a loading control. Changes in the levels of endogenous miR-29a and an untargeted control (miR-26a) were determined by Northern blotting. 5S rRNA served as a loading control. (**G**) Downregulation of endogenous truncated APC upon miR-29a overexpression. SW480 cells were transduced with lentiviruses encoding miR-29a or a control small RNA (siEGFP). Western and Northern assays were performed as in **F**. (**H**) Invariant concentration of endogenous wild-type APC upon miR-29a overexpression. HEK293 cells were transduced with lentiviruses encoding miR-29a or a control small RNA (siEGFP). Western and Northern assays were performed as in **F** except that Tubulin served as the loading control in the Western assay. (**I**) Duplexes expected for the miR-29a miREs base-paired with miR-29a. (**J**) Ribonucleoprotein immunoprecipitation (RIP) analysis of PTC-*APC* mRNA associated with Ago2 in SW480 cells upon miR-29a knockdown. SW480 cells used in **F** were transduced with a low amount of lentiviruses (MOI <0.3) expressing FLAG-tagged Ago2. Anti-FLAG RIP followed by qRT-PCR was performed to compare the binding of endogenous PTC-*APC* mRNAs to Ago2. The amount of Ago2-associated PTC-*APC* mRNA was normalized to *MYC*, an endogenous target of let-7c. The relative Ago2-RIP efficiency was calculated from the normalized amount of PTC-*APC* mRNA in the presence of a miR-29a decoy (TuD-29a) versus a control decoy (TuD-NC). (**K**) RIP analysis of PTC-*APC* mRNA associated with Ago2 in SW480 cells upon miR-29a overexpression. SW480 cells used in **G** were transduced with a low amount of lentiviruses (MOI <0.3) expressing FLAG-tagged Ago2. RIP assays were performed as in **J**. The relative Ago2-RIP efficiency was calculated from the normalized amount of Ago2-associated PTC-*APC* mRNA in the presence of miR-29a versus a control small RNA (siEGFP). (**L**) The 3′ UTR of *APC* mRNA contains no miR-29a miRE. The sequence of a full length *APC* 3′ UTR was cloned to the 3′ UTR of a firefly luciferase reporter. This reporter plasmid (pFL-APC-3′UTR) or a control plasmid (pFL) was co-transfected with a miR-29a mimic or a control small RNA (siNC) into HEK293 cells. The relative FL expression level was calculated from the normalized levels of firefly luciferase activity for pFL-APC-3′UTR versus pFL in the presence of the miR-29a mimic or siNC. (**M**) RIP analysis of ectopically expressed PTC-*APC* mRNA associated with Ago2 in HEK293 cells. A PTC-containing *APC* minigene plasmid with wild-type (APC-PTC1450) or mutant miR-29a miREs (APC-PTC1450-mut) was co-transfected with the miR-29a mimic or a control small RNA (siNC) into HEK293 cells that stably expressed FLAG-tagged Ago2. Anti-FLAG RIP followed by qRT-PCR was performed to compare the binding of the mRNAs to Ago2. The amount of Ago2-associated PTC-*APC* mRNA or its miRE mutant counterpart was normalized to *HOXD10*, an endogenous target of miR-10a. The relative Ago2-RIP efficiency was calculated from the normalized levels of Ago2-associated PTC-*APC* mRNA bearing wild-type (APC-PTC1450) or mutant (APC-PTC1450-mut) miR-29a miREs in the presence of the miR-29a mimic versus siNC.

The following figure supplements are available for figure 3:

**Figure supplement 1**. Duplexes expected for functional miRNAs base-paired with cognate miREs.

**Figure supplement 2**. Repressive activity of miR-29a miREs within the PTC-STOP region of *APC* minigenes bearing the natural *APC* 3′ UTR.

**Figure supplement 3**. Full functionality of miREs in the PTC-STOP region in the presence of 3′ UTR-dependent repression by miRNAs.

**Figure supplement 4**. Successful knockdown of endogenous miR-29a by TuD-29a.

**Figure supplement 5**. Confirmation of interaction between miR-29a and its miREs in *APC* mRNA.

panel), suggesting the upregulation observed for PTC-*APC* was specifically induced by the inhibition of miR-29a.

To test if miR-29a-mediated repression is specific for PTC-*APC* but not wild-type *APC* (WT-*APC*), we generated doxycycline (dox)-inducible miR-29a-overexpressing SW480 and HEK293 stable cell lines and examined the amount of endogenous APC protein after inducing miRNA expression for 3 days. HEK293 cells naturally express the full-length APC protein (***Figure 3D***) and low levels of miR-29a (***Figure 3H***, ***Figure 3—figure supplement 4B,D***). The amount of WT-APC protein remained unchanged after the induction of miR-29a expression to much higher levels in the cells (***Figure 3H***) because the miR-29a miREs are located within the PTC-STOP region but not the 3′ UTR of *APC* mRNA (***Figure 3A***). In contrast, SW480 cells that overexpress miR-29a produced a lower amount of truncated APC compared to cells that overexpressed a non-functional small RNA that did not affect the already high expression

levels of endogenous miR-29a (*Figure 3G*). These results support that miR-29a specifically represses the PTC-*APC* and does not impair the expression of WT-*APC*.

To further prove that the repressive effects of miR-29a on PTC-*APC* mRNA are direct, we performed ribonucleoprotein immunoprecipitation (RIP) assays to examine the association of PTC-*APC* mRNA with Ago2, the component of the RISC complex that directly binds the miRNA and its target mRNA. The level of endogenous PTC-*APC* mRNA associated with Ago2 showed a mild but reproducible decrease when miR-29a was knocked down (*Figure 3J*) and a significant increase upon miR-29a over-expression in SW480 cells (*Figure 3K*). The 3′ UTR of *APC* mRNA contains no miR-29a miREs, which was determined by comparing the expression of a luciferase reporter bearing a full-length *APC* 3′ UTR in the presence of miR-29a versus a negative control small RNA (siNC) (*Figure 3L*). Therefore, the changes in the levels of Ago2-associated PTC-*APC* mRNA in SW480 cells are most likely due to a direct effect of miR-29a targeting its miREs within the PTC-STOP region. Moreover, the binding efficiency of *APC*-PTC1450 mRNA bearing wild-type or mutant miREs to Ago2 was compared by co-transfecting HEK293 cells with the minigene construct and a miR-29a mimic or a control small RNA. In the presence of miR-29a, the *APC*-PTC1450 mRNA that harbors wild-type miR-29a miREs has a much stronger association with Ago2 than does its counterpart that contains the mutant miREs (*Figure 3M*). In addition, a mutant version of the miR-29a mimic that restored base-pairing of the 2–7 seed with the mutant miR-29a miRE also restored repression of *APC* mRNA (*Figure 3—figure supplement 5*), sup-porting the conclusion that repression of PTC-*APC* expression is achieved through the direct interac-tion of miR-29a with the miREs we mapped. Altogether, these observations represent novel evidence that the expression of a naturally occurring nonsense mRNA is selectively repressed by endogenous miRNAs in the cells.

## The repressive effects of miRNA-mediated surveillance and EJC-NMD are additive

Although *APC* provides an excellent case for studying miRNA-mediated surveillance, most of the reported nonsense mRNAs that harbor PTCs located upstream of the last exon should be EJC-NMD sensitive. We therefore sought to determine whether these EJC-NMD-competent transcripts are simultaneously subjected to miRNA-mediated surveillance. *BRCA1*, a tumor suppressor gene that is frequently mutated in breast cancer, was chosen for further investigation. Unlike *APC*, mutations that lead to the expression of a truncated version of BRCA1 are scattered along the ORF (*Castilla et al., 1994*); therefore, most *BRCA1* nonsense mutants contain introns downstream of the PTC and are EJC-NMD-sensitive (*Perrin-Vidoz et al., 2002*). We amplified the cDNA of the PTC-STOP region of one clinically identified *BRCA1* mutant (with a PTC at codon 526) (*Perrin-Vidoz et al., 2002*) and fused it to the 3′ end of a firefly luciferase coding region to create a chimera such that the original stop codon of the luciferase gene serves as a PTC in the fused reporter (*Figure 4A*). This reporter (pFL-BRCA1) has no downstream introns and harbors wild-type miREs; therefore, it is expected to be sensitive to miRNA only. pFL-BRCA1in is identical to pFL-BRCA1 except that two downstream introns were incorporated to render the transcript sensitive to EJC-NMD in addition to miRNA. The expression level of pFL-BRCA1in was significantly lower compared to pFL-BRCA1, indicating that EJC-NMD was contributing to the repressive activity (*Figure 4C*, left panel). When GW182, a key component of RISC, was knocked down >70% by small interfering RNAs (siRNAs) in HeLa-tTA cells (*Figure 4C*, right panel), the expres-sion level of pFL-BRCA1 was significantly increased, which indicates that miRNAs were indeed involved in the repression. More importantly, knockdown of GW182 also caused a comparable extent of upreg-ulation of pFL-BRCA1in, suggesting that miRNA-mediated repression of PTC-containing mRNAs is independent of the presence of downstream introns (*Figure 4C*, left panel). Thus, miRNA-mediated surveillance and EJC-NMD are not mutually exclusive, and the repressive effects caused by both mechanisms are additive.

Next, we identified several potential miREs between the PTC and the natural stop codon of *BRCA1* nonsense mutant mRNA by using the similar screening method for *APC* in HEK293 cells (data not shown). miREs targeted by miR-137 and miR-544 were mutated to create pFL-BRCA1-mut, which is not responsive to either EJC-NMD or miR-137 and miR-544, and pFL-BRCA1in-mut, which is insensitive to miR-137 and miR-544 but responsive to EJC-NMD (*Figure 4A,B*). As expected, the expression level of EJC-NMD-sensitive reporter pFL-BRCA1in-mut was significantly lower compared to pFL-BRCA1-mut (*Figure 4D*, compare black and light gray bars). When co-transfected with miR-137 or miR-544 in HEK293 cells (which do not naturally express these two miRNAs), the expression of the miRNA-specific reporter

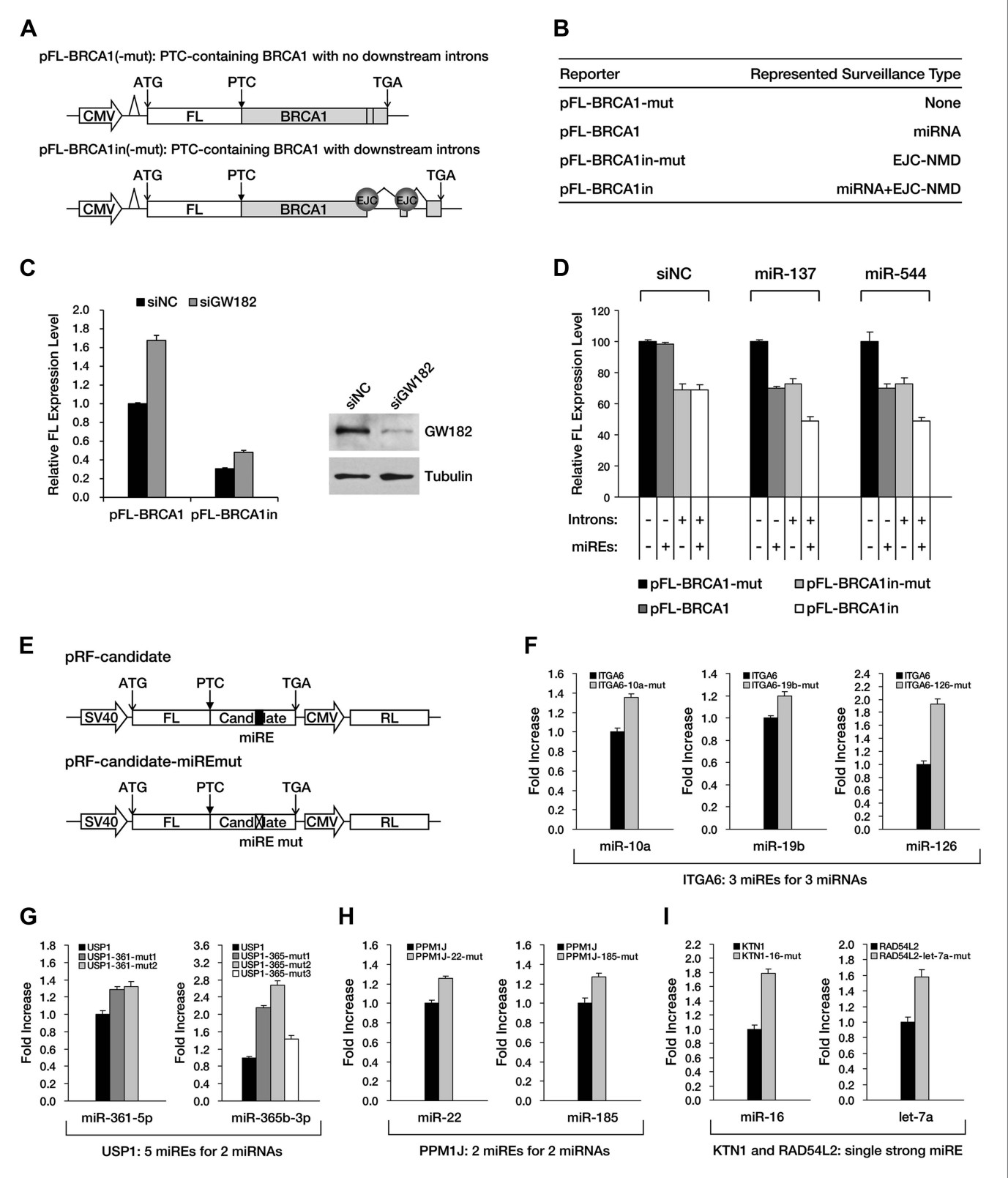

**Figure 4**. The repressive effects of miRNA-mediated surveillance and EJC-NMD are additive. (**A**) The reporter constructs used in **C** and **D**. pFL-BRCA1 and pFL-BRCA1-mut have no introns, whereas pFL-BRCA1in and pFL-BRCA1in-mut each harbor two introns downstream of PTC. The miREs for miR-137 and miR-544 were mutated to synonymous codons in pFL-BRCA1-mut and pFL-BRCA1in-mut. (**B**) Identification of the surveillance mechanism pertinent

*Figure 4. Continued on next page*

*Figure 4. Continued*

to each reporter. (**C**) Left panel: miRNA-mediated surveillance and EJC-NMD are not mutually exclusive. Knockdown of RISC core component GW182 alleviated the repression of *BRCA1* reporters, regardless of the presence of the downstream introns. The relative FL expression level represents the firefly/Renilla luciferase ratio in the presence of a control siRNA (siNC) versus an siRNA targeting GW182 (siGW182). Right panel: expression level of GW182 protein in HeLa-tTA cells treated with siNC or siGW182. Tubulin served as a loading control. (**D**) Additive effects of miRNA-mediated surveillance and EJC-NMD. Each reporter in **A** was co-transfected with miR-137, miR-544, or a control small RNA (siNC), together with a Renilla luciferase reporter into HEK293 cells. The relative FL expression level represents the firefly/Renilla luciferase ratio of each EJC-NMD- and/or miRNA-responsive reporter relative to pFL-BRCA1-mut. (**E**) Reporter constructs for miRE function validation of candidates identified from HCT-116 exome and RNA sequencing data. The PTC-STOP region of each candidate was fused to a firefly luciferase (FL) ORF in the same manner as for *APC* and *BRCA1*. A control Renilla luciferase (RL) reporter was expressed from a second promoter in the same construct (pRF-candidate). Each potential miRE identified from the screening was mutated to obtain a series of miRE mutant reporters (pRF-candidate-miREmut). (**F–I**) Experimental verification of functional miREs in the PTC-STOP region of selected candidates. The wild-type or miRE mutant version of each candidate reporter was co-transfected into HEK293 cells with a cognate miRNA mimic. The activity of each miRE (fold increase) was calculated from the normalized levels of firefly luciferase activity for pRF-candidate versus pRF-candidate-miREmut in the presence of the cognate miRNA mimic.

The following figure supplement is available for figure 4:

**Figure supplement 1**. Experimental verification of functional miREs in the PTC-STOP region of additional nonsense mutant mRNAs.

pFL-BRCA1 was significantly suppressed compared to pFL-BRCA1-mut, confirming the role of miRNA-mediated surveillance in repressing its target (*Figure 4D*, middle and right, compare black and dark gray bars). Importantly, a markedly stronger repression was observed for the reporter expected to be sensitive to both EJC-NMD and miRNA (pFL-BRCA1in) when co-transfected with miR-137 or miR-544 but not when co-transfected with a control small RNA (*Figure 4D*, compare black and white bars). This observation further validates that miRNA-mediated surveillance and EJC-NMD can act additively.

To determine if miRNA-mediated surveillance is a general mechanism that helps to eliminate nonsense transcripts, we sought to identify more potentially functional ORF miREs in other naturally occurring nonsense mRNAs. We performed exome sequencing of HCT-116 cells and identified 188 heterozygous nonsense mutations. Analysis of transcriptome data (*Djebali et al., 2012*) indicates that 47 of them were actively expressed in HCT-116 cells (*Supplementary file 2*). We chose 16 candidates that contain PTC-STOP regions longer than 400 nt for further experimental validation of miREs by using a similar strategy for *APC* and *BRCA1*. We found that 11 of the 16 candidates contained at least one functional miRE in the PTC-STOP regions (*Figure 4E–I*, *Figure 4—figure supplement 1*). Several of these candidates, such as *ITGA6*, *USP1*, and *PPM1J*, harbored multiple functional miREs that were responsive to different miRNAs (*Figure 4F–H*). Other candidates, such as *KTN1* and *RAD54L2*, contained at least one strong miRE (*Figure 4I*). As only a limited number of miRNAs have been screened by our luciferase assays, additional miREs may be found by increasing the screened miRNAs. Considering the well-known facts that multiple miRNAs can repress the same mRNA simultaneously and the effects of miREs are additive (*Doench and Sharp, 2004*), the presence of multiple active miREs within these PTC-STOP regions could result in a pronounced repressive effect. Together, our results strongly support the view that miRNA-mediated surveillance may serve as a general mechanism that downregulates various nonsense mRNAs in human cells (*Figure 5*).

## Discussion

In this study, we have described evidence that a PTC is sufficient to induce miRNA-mediated downregulation of nonsense mRNAs by unmasking miREs located between the PTC and the natural stop codon of the mRNA. We have also experimentally verified that nonsense mutants of *APC*, *BRCA1*, and a few other genes are subjected to miRNA-mediated surveillance in human cells. Our findings indicate that besides their established roles in regulating gene expression, miRNAs may serve as a novel surveillance system to reduce aberrant mRNAs bearing PTCs and their potentially harmful truncated protein products, thus functioning as an important supplement to other mRNA surveillance systems in mammalian cells.

Unlike classic NMD, miRNA-mediated surveillance is EJC-independent and recognizes its targets only if an miRE is embedded in the PTC-STOP region (*Figure 1A*) at a position as close as 10 nt downstream of the PTC (*Figure 2F,G*). The repressive influence of miRNA is enhanced by combining accelerated mRNA degradation and translational inhibition (*Figure 2C–E*). Furthermore, miRNA-mediated

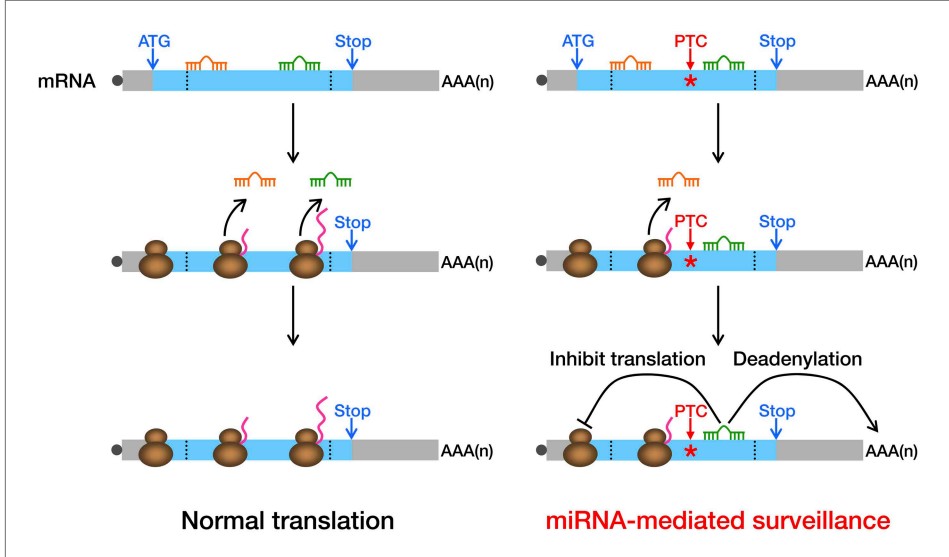

**Figure 5**. Model of miRNA-mediated surveillance system. The coding region of an mRNA may contain multiple potential miREs. Usually, miRNAs cannot stably bind to their cognate miREs that are embedded within the ORF of a normal transcript under active translation. However, upon nonsense mutation, the translating ribosome stalls at the PTC so that miREs downstream of the PTC are unmasked, triggering miRNA-mediated deadenylation and translational repression.

surveillance can work together with other surveillance mechanisms, such as EJC-NMD, in an additive manner (*Figure 4C,D*), which strengthens the repressive activity and expands the substrate spectrum of the cellular mRNA quality control system. An estimated one-third of genetic diseases are associated with truncated proteins produced from nonsense mRNAs (*Kuzmiak and Maquat, 2006*). Indeed, truncated BRCA1 has been shown to antagonize wild-type BRCA1 function in a dominant-negative manner (*Fan et al., 2001*; *Sylvain et al., 2002*), and several lines of evidence support the view that truncated APC contributes to colorectal cancer by causing spindle misalignment that eventually leads to chromosomal instability (*Fodde et al., 2001*; *Tighe et al., 2004*; *Quyn et al., 2010*). As predicted by a 2–7 seed match algorithm, the nonsense mutants of *APC* and *BRCA1* contain many potential conventional miREs in their PTC-STOP regions and several of them are experimentally validated in the cell lines, suggesting that these disease-causing mutants may be targets of miRNA-mediated surveillance. Recent studies have revealed that many unconventional miREs exist in animals (*Shin et al., 2010*; *Helwak et al., 2013*), raising the possibility that many more functional miREs may be present than would be predicted by a search algorithm based on the 2–7 seed match rule. In addition, targets for only a limited number of miRNAs were sought, which could also lead us to underestimate the frequency with which miREs are present in the PTC-STOP regions. The abundance of some endogenous miRNAs is already very high in HEK293 cells, which could lead them to falsely be scored as negative in our mimic-based screening. Such miRNAs could be identified by performing the screening in multiple cell lines that have different miRNA profiles. Moreover, the contribution of miRNAs to the overall level of repression may be even greater if we count the inhibitory effect of miRNAs on translation, which would not be evident in the RNA sequencing data (*MacArthur et al., 2012*).

Interestingly, we found that a luciferase reporter harboring the miR-29a miREs from the PTC-STOP region of *APC* nonsense mutant is efficiently repressed only in colon-derived SW480 cells but is not in kidney-derived HEK293 cells (*Figure 3—figure supplement 4B,D*), which correlates well with the relatively high abundance of miR-29a in SW480 cells and its undetectable level of expression in HEK293 cells (*Figure 3F–H*). This finding implies that a nonsense mRNA may have a very different fate in distinct tissues depending on which miRNAs are highly expressed. It is well established that the expression of many miRNAs are tissue- or cell type-specific and are dynamically regulated during cell differentiation or under different physiological conditions (*Houbaviy et al., 2003*; *Liu et al., 2004*; *Lu et al., 2005*;

*Marsit et al., 2006*; *Landgraf et al., 2007*; *Liang et al., 2007*). Some nonsense mRNAs may escape miRNA-mediated surveillance due to the lack of expression of certain miRNAs in some tissues or under some stress conditions, thereby increasing the risk of tissue-specific diseases.

In addition to many identified nonsense messages resulting from genomic mutations, evidence suggests that ~35% of alternatively spliced gene products contain PTCs (*Lewis et al., 2003*; *Wollerton et al., 2004*). Transcripts with retained introns, for example, are efficiently degraded by miRNAs, as shown by our BG reporter assays (*Figure 1D*). Considering that introns tend to be very long and rich in repeats, we believe many more transcripts with retained introns may be targeted by miRNAs. Consistent with this idea, more than 10% of Ago-CLIP reads map to introns (*Chi et al., 2009*; *Hafner et al., 2010*). Our computational analyses of transcriptome and published Ago-CLIP sequencing data from two human cell lines revealed that hundreds of PTC-containing transcripts generated by intron retention are indeed bound by Ago in the PTC-STOP regions (*Supplementary file 3*), which makes them attractive candidates of miRNA-mediated surveillance.

Taken together, we have uncovered a new role for miRNAs in mRNA quality control. This miRNA-mediated surveillance system acts in parallel with other systems, such as EJC-NMD, to provide extra protection for the cells against nonsense mutations. We have also demonstrated that *APC* and *BRCA1* nonsense mutant mRNAs are downregulated by multiple miRNAs that specifically target the PTC-STOP regions, thereby providing a feasible means to eliminate such deleterious nonsense transcripts without impairing their wild-type counterparts.

# Materials and methods

## Plasmid constructions

The rabbit β-globin coding region was amplified by PCR from the pBBB plasmid (*Shyu et al., 1989*) and placed downstream of a tet-off promoter to generate the plasmid TBG. LastEx-PTC was constructed by mutating one nucleotide at codon 121 to introduce a PTC in TBG. The LastEx-L7 and LastEx-PTC-L7 plasmids were constructed by inserting one let-7a miRE originating from human *LIN-28* (GCACAGCCTA TTGAACTACCTCA) in-frame into TBG and LastEx-PTC. hp-LastEx-L7 is identical to LastEx-L7 except for the presence of a stable hairpin (GGGGCGCGTGGTGGCGGCTGCAGCCGCCACCACGCGCCCC) in the β-globin 5′ UTR 29 nt upstream of the initiation codon. A Renilla luciferase ORF was fused to the 5′ end of BG ORF to create a RL-BG chimeric reporter, and then the stable hairpin was placed in the 5′ UTR at the same position as hp-LastEx-L7 to generate hp-RL-BG. Two nucleotides within the let-7a miRE seed region were mutated without changing the amino acids to generate LastEx-PTC-L7M and hp-LastEx-L7M. Two nucleotides of the 5′ splicing site and three nucleotides of the 3′ splicing site within the final intron of TBG or LastEx-L7 were mutated to create TBG-IR or TBG-IR-L7. Two nucleotides within the let-7a miRE seed region were mutated without changing the amino acids to generate TBG-IR-L7M. For the second set of TBG plasmids containing a miR-21 miRE, all the other designs are identical to the TBG plasmids with a let-7a miRE except for two differences: (1) the PTC site was introduced at codon 116, and (2) an artificial miR-21 miRE complementary to miR-21 with three centered mismatches (TCAACATCAGAGAGATAAGCTA) was inserted in-frame into TBG. The plasmid 3′UTR-L7 has been described (*Wu et al., 2006*). The let-7a miRE between the NheI and XbaI sites was replaced by a miR-21 miRE to generate 3′UTR-21. PTC102, PTC102-L7, and PTC102-L7M are identical to LastEx-PTC, LastEx-PTC-L7, and LastEx-PTC-L7M, respectively, except that the PTC site was introduced at codon 102. The let-7a miRE or its mutant in PTC102-L7 or PTC102-L7M was replaced with a miR-21 miRE or its mutant to generate PTC102-21 and PTC102-21M. Two tandem miR-125b miREs (GGTATCACAAGTTACAATC TCAGGGATAGCCAAGGTATCACAAGTTACAATCTCAGGGATAGCCAATTCTTAAT) were inserted between the EcoRI and XbaI sites 59 nt downstream of the firefly luciferase ORF with modified 3′ UTR sequences containing an additional stop codon to generate TAA-2E. The original stop codon of the luciferase gene was disrupted by a point mutation to obtain TCA-2E. One miR-125b miRE was inserted in-frame into a modified firefly luciferase plasmid at different positions before or after the PTC to generate a series of miRE-containing luciferase plasmids. The *APC* minigene plasmid was constructed by placing an HA-tagged *APC* ORF with or without its natural 3′ UTR downstream of the CMV promoter. An HA-tagged *EGFP* ORF was driven by an SV40 promoter from the same plasmid backbone. PTCs and the corresponding miRE mutations were generated by site-directed mutagenesis. miRE screening plasmids for *APC*, *BRCA1*, and all of the selected candidates from HCT-116 cells were constructed by fusing corresponding PTC-STOP regions to the 3′ end of a firefly luciferase ORF. miRE validation plasmids

were constructed either by inserting ~30 nt sequences that surround predicted miRE seeds into the 3′ UTR of a firefly luciferase reporter gene or by fusing the whole PTC-STOP region of each candidate with the mutated miRE seed to the 3′ end of the firefly luciferase ORF. The last 738 nt of *APC* coding region (PTC-STOP region) plus the entire 3′ UTR were fused to the 3′ end of a firefly luciferase ORF to generate pFL-APC-WT. The miR-29a miREs and miR-135b miRE were disrupted individually or together by site-directed mutagenesis to generate pFL-APC-29M, pFL-APC-135M, and pFL-APC-29M+125M. The sequence of the *APC* natural 3′ UTR was cloned downstream of a firefly luciferase gene to generate pFL-APC-3′UTR. Mature miR-29a or a negative control small RNA sequence siEGFP (AACTTCAGGGTCAGCTTGCCG) was cloned into the inducible TRIPZ lentiviral shRNA vector to generate miRNA-overexpression lentiviruses. A miRNA decoy sequence (TuD-NC or TuD-29a) was cloned into the GIPZ constitutive lentiviral vector to generate miRNA-knockdown lentiviruses. pFL-BRCA1 is identical to the *BRCA1* miRE screening plasmid. pFL-BRCA1in harbors two introns downstream of the luciferase stop codon. Functional miR-137 and miR-544 miREs within the *BRCA1* PTC-STOP region were subjected to a synonymous mutation to generate pFL-BRCA1-mut and pFL-BRCA1in-mut.

## Cell culture and stable cell lines
HEK293, 293T, HCT-116, and SW480 cells were purchased from ATCC (Manassas, Virgina, USA). HeLa-tTA cells were purchased from Clontech (Mountain View, California, USA). All cells were maintained in Dulbecco's modified Eagle's medium supplemented with 10% fetal bovine serum (Invitrogen, Carlsbad, California, USA). To produce the lentiviruses, 293T cells were transfected with a virus vector encoding the expression cassette as well as the VSVG and ΔR8.91 plasmids. Viruses were harvested at 48 and 72 hr post-transfection. HEK293, HCT-116, and SW480 stable cell lines were generated by transduction with lentiviruses in the presence of 8 μg/ml of polybrene overnight, followed by a 1-week puromycin and/or hygromycin selection.

## RNA interfering to deplete RISC core component
Endogenous GW182 of HeLa-tTA cells was depleted using protocols described elsewhere (*Piao et al., 2010*). siRNAs were synthesized by GenePharma (Shanghai, China) and the mature sequences are as follows: siNC: UUCUCCGAACGUGUCACGUUU; siGW182: UUGAGCACGGAGAUUAGGCUG.

## Deadenylation and decay assays
HeLa-tTA cells were plated on 35-mm plates 1 day before transfection in DMEM containing 20 ng/ml tet. In total, 350 ng of the BG reporter plasmid was used for transfection. The transcription of BG mRNA was induced by removing tet 12 hr after transfection. After 3 hr of induction, tet was added to a final concentration of 1 μg/ml to block the transcription of BG. Cytoplasmic RNA was then isolated at various time intervals. Equal amounts of RNA were treated with RNase H in the presence of an oligodeoxynucleotide (GGTTGTCCAGGTGACTCAGACCCTC) complementary to codons 74–81 within the BG coding region. For BG reporters with a retained last intron, RNA samples were treated with RNase H in the presence of an oligodeoxynucleotide (CAGTGTATATCATTGTAACCATAAA) complementary to a region within the last intron which is 81 nt upstream of the final exon. The digested RNA samples were then analyzed by electrophoresis (5.5% PAGE with 8 M urea) and Northern blotting as previously described (*Wu et al., 2006*; *Wu and Belasco, 2008a*). For measuring BG mRNA half-life, a constitutively transcribed *EGFP* mRNA was co-transfected with BG plasmids.

## Luciferase assays
In the miR-125b repression assays, HEK293 cells cultured in a 12-well plate were transfected with a firefly luciferase reporter (TAA-2E, TCA-2E, or a series of firefly luciferase reporters with one miR-125b miRE embedded before or after the PTC, 10 ng), pRL (10 ng), and a plasmid encoding or not encoding miR-125b (pMIR125b or pMIR125bΔ, respectively; 480 ng). In the miRE screening assays of *APC*, *BRCA1*, and computationally identified candidates from HCT-116 cells, HEK293 cells cultured in a 24-well plate were transfected with a firefly luciferase reporter containing the corresponding PTC-STOP region of the selected candidate (20 ng), pRL (10 ng), and 10 pmol of a synthetic miRNA mimic. In the miRE validation assays, HEK293 cells cultured in a 24-well plate were transfected with a vector encoding both firefly and Renilla luciferase (10 ng) and 10 pmol of the cognate miRNA mimics. In the PTC-STOP and 3′ UTR miRE additive effect assay, HEK293 cells cultured in a 24-well plate were transfected with a firefly luciferase reporter having a partial ORF and the entire 3′ UTR of *APC* fused to the 3′ end

of the luciferase ORF (pFL-APC-WT or its miRE mutant counterparts, 10 ng), pRL (10 ng), and 10 pmol of siNC or miR-29a and miR-135b mimic mixture. To examine potential miR-29a miREs in the natural *APC* 3′ UTR, HEK293 cells cultured in a 24-well plate were transfected with a normal firefly luciferase reporter (pFL, 10 ng) or a reporter bearing a full length *APC* 3′ UTR sequence in the luciferase 3′ UTR (pFL-APC-3′UTR, 10 ng), pRL (10 ng), and 10 pmol of siNC or miR-29a mimics. In the miRNA-mediated surveillance and EJC-NMD additive effect assay, HEK293 cells cultured in a 24-well plate were transfected with a firefly luciferase reporter harboring the *BRCA1* PTC-STOP region with or without introns (pFL-BRCA1 or its miRE mutant version pFL-BRCA1-mut, 20 ng; pFL-BRCA1in or its miRE mutant version pFL-BRCA1in-mut, 33.3 ng), pRL (10 ng), and 10 pmol of siNC or miRNA mimics. In the GW182 knockdown assay, siRNA-treated HeLa-tTA cells cultured in a 24-well plate were transfected with pFL-BRCA1 (20 ng) or pFL-BRCA1in (33.3 ng), together with 10 ng pRL. In the tissue-specific repression of miRNA-mediated surveillance and miR-29a knockdown assays, HEK293 cells, SW480 cells, and TuD-NC- or TuD-29a-overexpressing SW480 cells cultured in a 24-well plate were transfected with a vector encoding both a firefly and a Renilla luciferase reporter (20 ng) with the *APC* miR-29a miREs or the mutant form in the 3′ UTR of the firefly luciferase gene. In all luciferase assays, values represent means±SD from at least three independent experiments.

## Real-time quantitative PCR

In total, 2 µg cytoplasmic RNA isolated from HEK293 cells was treated with 1 U DNase I (Fermentas, Burlington, Ontario, Canada) and was then reverse transcribed using M-MLV (TAKARA, Otsu, Shiga, Japan), according to the manufacturer's instructions. Real-time PCR was performed on a StepOnePlus real-time PCR system (Applied Biosystems, Foster City, California, USA) with Power SYBR Green PCR Master Mix (Applied Biosystems, Foster City, California, USA). The PCR mixtures were heated to 95°C for 10 min and then subjected to 40 amplification cycles (15 s at 95°C, 1 min at 60°C).

## Immunoblot assays

For protein expression from the *APC* minigene construct, 1 µg of the minigene vector encoding both HA-APC and HA-EGFP and 100 pmol of the synthetic miRNA mimics were transfected into HEK293 cells cultured in 6-well plates using Lipofectamine 2000 (Invitrogen, Carlsbad, California, USA). After 36 hr, protein extracts of transfected cells were separated on 7.5% polyacrylamide-SDS gels by electrophoresis. For blotting full length APC, 6% gels were used. After electrophoresis, the gels were cut according to the pre-stained protein size marker. The portion that contained the internal control (HA-EGFP, GAPDH, or Tubulin) was transferred to a nitrocellulose membrane using standard transfer buffer (25 mM Tris, 192 mM glycine, 10% methanol) at 200 mA for 2 hr. The portion that contained the APC protein was transferred with a different transfer buffer (50 mM Tris, 380 mM glycine, 0.2% SDS, 5% methanol) at 25 V for 20 hr. After blocking with 5% non-fat milk in phosphate-buffered saline (PBS) containing 0.05% Tween-20 (PBST) for 30 min at room temperature, the blot was probed for 1 hr at room temperature with anti-HA antibodies (1:3000; ImB, Shanghai, China), anti-APC antibodies (1:200; Millipore, Billerica, Massachusetts, USA), anti-GAPDH antibodies (1:3000; Bioworld, St. Louis Park, Minnesota, USA), or anti-α-Tubulin antibodies (1:5000; Sigma-Aldrich, St. Louis, Missouri, USA) diluted in PBST buffer with 1% BSA, then incubated with a secondary antibody conjugated to horseradish peroxidase (1:10,000; Jackson ImmunoResearch Laboratories, West Grove, Pennsylvania, USA) in 5% non-fat milk, and detected with an Immun-Star HRP chemiluminescence kit (Bio-Rad, Hercules, California, USA). For validation of GW182 knockdown in HeLa-tTA cells, an anti-GW182 antibody (1:1000; MBL, Nagoya, Japan) was used.

## Ribonucleoprotein immunoprecipitation

For RIP assays of ectopically expressed PTC-*APC* mRNAs, HEK293 cells that stably express FLAG-tagged Ago2 were plated in 60-mm plates 1 day before transfection. A total of 2 µg of APC-PTC1450 or APC-PTC1450-mut plasmid with 150 pmol of siNC or miR-29a mimic was transfected into the cells. Then 36 hr after transfection, cells were trypsinized, collected, and washed with 10 ml PBS twice. The pelleted cells were lysed in 200 µl PLB buffer (100 mM KCl, 5 mM MgCl$_2$, 10 mM HEPES, 0.5% NP-40, 1 mM DTT, 100 U/ml RNase inhibitor). The cleared lysate was incubated with anti-FLAG affinity gel (Sigma-Aldrich, St. Louis, Missouri, USA) for 1 hr. Beads were washed with 1 ml NT-2 buffer (50 mM

Tris, 150 mM NaCl, 1 mM MgCl$_2$, 0.05% NP-40) five times. Washed beads were re-suspended in 250 µl NT-2 buffer and RNA was isolated using Trizol LS Reagent (Ambion, Austin, Texas, USA) and analyzed by qRT-PCR. For RIP assays of endogenous PTC-*APC* mRNAs, SW480 cells that stably express FLAG-tagged Ago2 were cultured in one 100-mm plate and were used for RIP assay after they reached 80% confluency.

## Bioinformatics analysis strategy

### SNV calling
Genomic DNA was extracted from HCT-116 cells using the Genomic DNA Extraction kit (TIANGEN, Beijing, China). Exon-captured libraries were constructed by Genergy Biotechnology (Shanghai, China). Sequencing was conducted using the Illumina Hiseq2000 to produce pair-end reads. The exome sequencing data (24 × coverage) for HCT-116 cells have been deposited in the Short Read Archive (SRA accession number: SRX528176). Raw exome sequencing data were mapped to the human genome (hg19) using BWA with five mismatches allowed per uniquely aligned read, and duplicate reads were marked using Picard. The alignments were then processed to generate SNV calls as a standard workflow by GATK. A human SNV dataset was downloaded from the GATK resource bundle (ftp.broadinstitute.org). SNVs that lead to premature termination were designated as PTCs and were used in further analysis. Genes with a PTC/WT allele reads ratio between 0.7 and 1.3 were considered to be heterozygous.

### Allele-specific expression analysis of heterozygous PTC variants
Published transcriptome sequencing data of the HCT-116 cell line (GEO accession number: GSM958749) were used to quantify the relative expression levels for the identified PTC-containing genes. Raw RNA sequencing data were mapped to heterozygous PTC loci using Tophat. Duplicate-removed alignments were processed to generate PTC calls using Samtools. Functional annotation of PTCs was performed with reference to GRCh37.70 using snpEff. For transcripts with multiple isoforms, the one with the longest PTC-STOP region was annotated for downstream analysis. PTC-containing transcripts were subdivided into EJC-NMD-sensitive and/or miRNA-mediated surveillance-sensitive groups: if a PTC site was found more than 50 nt upstream of the final exon–exon junction in a transcript, it was regarded as an EJC-NMD- and miRNA-mediated surveillance-sensitive candidate; otherwise the transcript was considered a miRNA-mediated surveillance-sensitive candidate.

### miRE prediction
miRNA target site prediction was performed within the PTC-STOP region of the selected candidates. The small RNA sequencing data have been deposited in the Short Read Archive (SRA accession numbers: SRX528179 for HCT-116 cells; SRX528184 for HeLa cells; SRX528182 for HEK293 cells). The top 150 expressed miRNAs were included in the miRE prediction. miRBase v20 was used as the reference database. miRE prediction was based on the 2–7 seed match rule: if the 5' 2–7 seed region of the miRNA formed Watson–Crick base pairs with sequences within regions at least 10 nt downstream of the PTC of the candidate mRNA without mismatches, this mRNA was considered to harbor one miRE for the selected miRNA.

### Intron retention prediction
Raw RNA sequencing data of HeLa (SRA accession number: SRX528183) and HEK293 cells (SRA accession number: SRX528181) were processed through the standard Tophat and Cufflinks pipeline to generate transcriptome profiles. The isoform with the highest transcript abundance in each gene family was chosen as the constitutive isoform. Intron-retention isoforms with an intron retention level >0.05 were then found by using MATS accordingly. For all the PTC-causing intron-retention isoforms, Ago-CLIP hits (http://starbase.sysu.edu.cn/download.php) were searched for in their PTC-STOP regions. miREs were predicted based on the 2–7 seed match rule as described above.

## Acknowledgements
We thank Joel Belasco and Yixian Zheng for critical reading and correction of the manuscript. We thank Hideo Iba for providing TuD-NC plasmid and Degui Chen for *USP1* cDNA plasmid. This work was supported by grants from the Ministry of Science and Technology of China (2011CB811303, 2012CB910802), National Natural Science Foundation of China (31270841, 30970618), and the Chinese Academy of Sciences (2008OHTP01).

## Additional information

### Funding

| Funder | Grant reference number | Author |
|---|---|---|
| Ministry of Science and Technology of the People's Republic of China | 31270841, 30970618 | Ligang Wu |
| National Natural Science Foundation of China | 2011CB811303, 2012CB910802 | Ligang Wu |

The funders had no role in study design, data collection and interpretation, or the decision to submit the work for publication.

### Author contributions

YZ, Conception and design, Acquisition of data, Analysis and interpretation of data, Drafting or revising the article; JL, Acquisition of data, Analysis and interpretation of data, Drafting or revising the article; BX, SH, XZ, Acquisition of data, Drafting or revising the article; LW, Conception and design, Analysis and interpretation of data, Drafting or revising the article

## Additional files

### Supplementary files

• Supplementary file 1. Screening results of functional miREs in the PTC-STOP region of *APC* mRNA. Reporter constructs and the strategy for miRE screening were described in *Figure 3—figure supplement 1A*. In total, 45 miRNAs based on their general abundance in human tissues and the predicted thermal stability of the duplex they may form with an *APC* miRE were chosen for the screening. The repression ratio for each miRNA was calculated from normalized levels of firefly luciferase activity in the presence of a control small RNA (siEGFP) versus a miRNA mimic.

• Supplementary file 2. Transcriptome-wide identification of nonsense mRNAs caused by point mutation in HCT-116 cells. Exome sequencing was performed to identify all PTC-containing mutants in the genome of HCT-116 cells. Genes with the PTC/WT allele reads ratio between 0.7 and 1.3 were considered to be heterozygous. Transcriptome sequencing data were then analyzed to evaluate the effect of these heterozygous PTCs on mRNA stability. Sequencing reads that specifically mapped to the PTC and wild-type allele were counted and used to calculate the relative concentration of the respective transcripts. In total, 47 heterozygous PTC-containing mRNAs predicted from the genomic sequence were actively expressed in HCT-116 cells, as shown in the list. Candidates were subdivided into EJC-NMD-sensitive and/or miRNA-mediated surveillance-sensitive groups according to the position of the PTC. miREs in the PTC-STOP region of each candidate were predicted based on the 2–7 seed match rule and the top 150 expressed miRNAs in HCT-116 cells. Candidates highlighted in yellow were selected for experimental validation. Candidates that contain at least one experimentally verified miRE are shown in red font.

• Supplementary file 3. Transcriptome-wide identification of nonsense mRNAs caused by intron retention in HEK293 and HeLa cells. The isoform with the highest transcript abundance in each gene family was chosen as the constitutive isoform. All intron-retention (IR) isoforms with an intron retention level >0.05 were then found by using MATS. Only PTC-causing IR candidates with Ago-CLIP hits in their PTC-STOP regions were listed. miREs were predicted in the PTC-STOP region using the 2–7 seed match rule. Candidates were subdivided into EJC-NMD-sensitive and/or miRNA-mediated surveillance-sensitive groups.

### Major datasets

The following datasets were generated:

| Author(s) | Year | Dataset title | Dataset ID and/or URL | Database, license, and accessibility information |
|---|---|---|---|---|
| Zhao Y, Lin J, Xu B, Hu S, Zhang X, Wu L | 2014 | Data from: MicroRNA-mediated repression of nonsense mRNAs | http://www.ncbi.nlm.nih.gov/sra/SRX528176 | Publicly available at NCBI Short Read Archive. |

| Zhao Y, Lin J, Xu B, Hu S, Zhang X, Wu L | 2014 | Data from: MicroRNA-mediated repression of nonsense mRNAs | http://www.ncbi.nlm.nih.gov/sra/SRX528183 | Publicly available at NCBI Short Read Archive. |
|---|---|---|---|---|
| Zhao Y, Lin J, Xu B, Hu S, Zhang X, Wu L | 2014 | Data from: MicroRNA-mediated repression of nonsense mRNAs | http://www.ncbi.nlm.nih.gov/sra/SRX528181 | Publicly available at NCBI Short Read Archive. |
| Zhao Y, Lin J, Xu B, Hu S, Zhang X, Wu L | 2014 | Data from: MicroRNA-mediated repression of nonsense mRNAs | http://www.ncbi.nlm.nih.gov/sra/SRX528179 | Publicly available at NCBI Short Read Archive. |
| Zhao Y, Lin J, Xu B, Hu S, Zhang X, Wu L | 2014 | Data from: MicroRNA-mediated repression of nonsense mRNAs | http://www.ncbi.nlm.nih.gov/sra/SRX528184 | Publicly available at NCBI Short Read Archive. |
| Zhao Y, Lin J, Xu B, Hu S, Zhang X, Wu L | 2014 | Data from: MicroRNA-mediated repression of nonsense mRNAs | http://www.ncbi.nlm.nih.gov/sra/SRX528182 | Publicly available at NCBI Short Read Archive. |

The following previously published dataset was used:

| Author(s) | Year | Dataset title | Dataset ID and/or URL | Database, license, and accessibility information |
|---|---|---|---|---|
| Djebali S, Davis CA, Merkel A, Dobin A, et al. | 2012 | Data from: Landscape of transcription in human cells | http://www.ncbi.nlm.nih.gov/geo/query/acc.cgi?acc=GSE33480 | Publicly available at NCBI Gene Expression Omnibus. |

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
