## [Decision Letter]

Thank you for sending your work entitled “MicroRNA-mediated repression of nonsense mRNAs” for consideration at *eLife*. Your article has been favorably evaluated by James Manley (Senior editor) and 2 reviewers, one of whom, Nick Proudfoot, is a member of our Board of Reviewing Editors.

The Reviewing editor and the other reviewer discussed their comments before we reached this decision, and the Reviewing editor has assembled the following comments to help you prepare a revised submission.

This manuscript describes an interesting new mechanism of nonsense-mediated decay (NMD). In contrast to NMD mediated by the exon junction complex (EJC), this process is mediated by miRNAs. Using several reporter assays, the authors show that when a premature termination codon (PTC) is present, the region of the ORF downstream becomes accessible to miRNA-mediated repression. They go on to investigate a naturally occurring example of NMD in the APC gene, and find that this is mediated by several different miRNAs that bind to the region between the PTC and the stop codon. They then use reporters based on BRCA1 to show that the effects of miRNA-mediated and EJC-mediated NMD are additive. Finally, the authors identify PTC mutations in number of genes and show miRNA-mediated NMD in a reporter construct for several of these, suggesting that this mechanism may be widespread.

This is an interesting manuscript that describes an important novel function for miRNAs. The data are clear and generally support the authors' conclusions. However some points need addressing in a revised manuscript.

Figure 1:

1) That the Northern blot shortening product is truly poly(A) shortening should be confirmed using RNase H and oligo(dT) targeting. Also it would be important to determine whether β-globin mRNA levels decrease as a consequence of this.

2) It is stated that the natural sequence of β-globin mRNA does not contain any miRNA target sites. Does this apply to the region between the PTC and original stop codon in all their constructs, even the ones with the retained intron or with the PTC in exon 2? Given the prevalence of potential seed matches for the numerous human miRNAs, it seems quite unlikely that there would be no targeting in these longer regions. Deadenylation in the intron-retained β-globin mRNA without the let-7 target site (Figure 1) does seem to be more rapid than in the other reporters, which could be explained by endogenous miRNA targeting.

3) The 5' UTR hairpin should be directly shown to block β-globin translation.

4) The original paper, Wollerton et al Mol Cell. 2004 13, 91-100 should also be cited for AS causing NMD. This was the original study I think.

Figure 2:

In Figure 2, both protein and mRNA levels are lower from the reporter without a PTC than from the reporter with one. Do the authors know why this is? It's possible that the presence of a C-terminal extension on luciferase might affect its translation or protein stability, which could influence the results of these assays.

Figure 3:

1) The screening approach relies on overexpression of miRNAs. It is probable that some of the miRNAs tested will be expressed in HEK293, and that there will be some variability in their expression. This would be likely to affect the results of the screening, and needs discussion with reference to known HEK293 miRNA expression levels.

2) These experiments while elegant need further data on direct effects of miR-29 as there is a possibility here of indirect effects. I think evidence that miR-29 really associates with the APC PTC mutant RNA in vivo needs to be obtained by Ago pull down of the specific mRNA. Alternatively an Ago CLIP analysis might be useful.

Figure 4:

The authors do not consider that in the context of an endogenous PTC-containing mRNA, it is likely that the 3' UTR will be targeted by miRNAs. What would the effects of this targeting be, and would the unmasking of additional sites upstream of the original stop codon make a significant addition to existing 3' UTR-mediated miRNA repression? The constructs used to assess APC and BRCA1 NMD lack the endogenous 3' UTR; it would be important to include this in at least some of the experiments to assess how this influences miRNA-mediated NMD.

---

## [Author Response]

Figure 1*:*

*1) That the Northern blot shortening product is truly poly(A) shortening should be confirmed using RNase H and oligo(dT) targeting. Also it would be important to determine whether β-globin mRNA levels decrease as a consequence of this*.

We have performed the experiments suggested by the reviewers and incorporated the data into Figure 1 and Figure 1—figure supplement 1. To identify at which end of β-globin (BG) mRNA the shortening occurs, the RNA samples were treated with RNase H and an oligodeoxynucleotide complementary to the interior of the BG reporter mRNA (codons 74-81) to generate 5’ and 3’ fragments, which were then separated by electrophoresis on a denaturing polyacrylamide gel and detected by Northern blotting. In addition to the Northern blot examination of the 3’ cleavage products of BG mRNAs shown previously in Figure 1, we have now also included Northern blot analysis of the 5’ fragments in Figure 1 (bottom portion) and demonstrated that shortening at the 5’ end of BG mRNA is undetectable during the entire period, in contrast to the shortening of the 3’ fragments that results from removal of the poly(A) tails. Moreover, treatment of the RNA samples with oligo(dT) and RNase H caused the BG 3’ fragments that previously appeared as diffuse bands after electrophoresis to migrate uniformly to a position corresponding to fully deadenylated mRNA (Figure 1), which constitutes additional evidence for 3’ poly(A) tail shortening.

We have also measured the decay rate of the BG mRNAs by monitoring the declining concentration of each as a function of time after transcriptional arrest by a method described previously (Wu and Belasco, *Methods in Enzymology*, 2008). To more accurately quantify the half-life of BG mRNA (wild-type BG mRNA has an intrinsically long half-life of >12 hr), we collected RNA samples at later time points than we did previously when monitoring mRNA deadenylation. The results demonstrate that, as a consequence of accelerated deadenylation, the BG mRNA bearing both the let-7a miRE and the upstream PTC decayed much faster than counterparts that lacked the PTC or the let-7a miRE or bore the mutant let-7a miRE, its half-life decreasing from >6 hr to <3 hr. LastEx-L7 decayed slightly faster than the control BG mRNAs (TBG and LastEx-PTC), which suggests that the miRE located in the ORF may still have residual activity. Importantly, the presence of a stable hairpin in the 5’ UTR (hp-LastEx-L7) significantly increased the decay rate of LastEx-L7 mRNA, with the half-life decreasing from >6 hr to <3 hr. These findings are consistent with the deadenylation assays shown in Figure 1, which suggest that translating ribosomes may indeed have interfered with miRNA-RISC binding to the miREs located in the ORF and thus masked their repressive function. The representative results from multiple experiments are now included as Figure 1—figure supplement 1.

*2) It is stated that the natural sequence of β-globin mRNA does not contain any miRNA target sites. Does this apply to the region between the PTC and original stop codon in all their constructs, even the ones with the retained intron or with the PTC in exon 2? Given the prevalence of potential seed matches for the numerous human miRNAs, it seems quite unlikely that there would be no targeting in these longer regions. Deadenylation in the intron-retained β-globin mRNA without the let-7 target site (*Figure 1*) does seem to be more rapid than in the other reporters, which could be explained by endogenous miRNA targeting*.

We thank the reviewers for pointing this out. We have analyzed all of the BG constructs for potential miREs and added this information to the manuscript. None of the BG constructs shown in Figure 1 except the ones that retained the last intron have predicted miREs in their PTC-STOP regions, as judged by using the 2–7 seed match rule and the 100 most abundant miRNAs in HeLa cells. This finding is not surprising since these regions are very short (84 nt in LastEx-PTC and 138 nt inPTC102). However, retention of the last intron in TBG-IR would introduce a 570-nt PTC-STOP region that contains multiple predicted miREs (listed in the table below). Some of the predicted miREs are targets of highly expressed endogenous miRNAs in HeLa cells, such as miR-10a/b and miR-17/20a. This may explain the slightly accelerated deadenylation rate of the TBG-IR reporter, which is consistent with our model that non-functional miREs embedded in retained introns can become functional miREs in the presence of an upstream PTC.

Author response table: Predicted miREs within the last intron of BG mRNAmiRNARegionmiRE start positionmiRE seedhsa-miR-494-3plast intron138TGTTTChsa-miR-186-5plast intron141TTCTTT***hsa-miR-17-5p***last intron147CACTTT***hsa-miR-20a-5p***last intron147CACTTThsa-miR-186-5plast intron186TTCTTT***hsa-miR-17-5p***last intron251CACTTT***hsa-miR-20a-5p***last intron251CACTTT***hsa-miR-17-5p***last intron292CACTTT***hsa-miR-20a-5p***last intron292CACTTThsa-miR-125b-5plast intron311TCAGGGhsa-miR-125a-5plast intron311TCAGGG***hsa-miR-10a-5p***last intron312CAGGGT***hsa-miR-10b-5p***last intron312CAGGGThsa-miR-374a-5plast intron320ATTATAhsa-miR-374b-5plast intron320ATTATAhsa-miR-423-5plast intron530CCCCTC

*3) The 5' UTR hairpin should be directly shown to block β-globin translation*.

Evidence that a large stem-loop structure in the 5’ UTR blocks translation of β-globin mRNA is now included in Figure 1—figure supplement 3. We fused the BG reporter in-frame to a cDNA fragment encoding Renilla luciferase, which allowed us to measure protein levels more accurately. The concentration of the mRNA in which a large stem-loop structure was inserted into the 5’ UTR was comparable to that of the corresponding construct without a hairpin, whereas the amount of protein produced by the mRNA with the stem-loop was only 2% as high, suggesting very efficient blocking of mRNA translation by the stem-loop in the 5’ UTR. In addition, previous reports of using the same stem-loop in the 5’ UTR to block translation initiation (Chen et al., Mol Cell Biol, 1995; Wu et al., Proc Natl Acad Sci U S A, 2006) are now cited.

*4) The original paper, Wollerton et al Mol Cell. 2004 13, 91-100 should also be cited for AS causing NMD. This was the original study I think*.

We now cite the paper as suggested.

Figure 2*:*

*In*
Figure 2*, both protein and mRNA levels are lower from the reporter without a PTC than from the reporter with one. Do the authors know why this is? It's possible that the presence of a C-terminal extension on luciferase might affect its translation or protein stability, which could influence the results of these assays*.

It seems that the similar appearance of the bar graphs in Figure 2 may have caused confusion about what is shown in panel E. The values graphed in Figure 2 are repression ratios, which were calculated by dividing the FL protein or mRNA levels in the absence and in the presence of miRNA (miR-125b). A value >1 indicates a repressive effect, so the values for TAA-2E (the one with a PTC) should be higher than those of TCA-2E (the one without a PTC, which has a repression ratio of ∼1, indicating no repressive effect). We now have added a horizontal red line at a ratio of 1.0 and a clearer explanation in the figure legend to help readers interpret the data.

Figure 3*:*

*1) The screening approach relies on overexpression of miRNAs. It is probable that some of the miRNAs tested will be expressed in HEK293, and that there will be some variability in their expression. This would be likely to affect the results of the screening, and needs discussion with reference to known HEK293 miRNA expression levels*.

We agree with the reviewers’ concerns. Most of the miRNA candidates that had a strong repressive effect in the screen with the PTC-*APC* reporter ([Supplementary-material SD1-data]), such as miR-138, miR-29a, miR-127-5p and miR-378, have very low or undetectable expression levels in HEK293 cells (in-house sequencing data). Furthermore, other endogenous miRNAs, such as miR-21 and miR-103a-3p, have multiple predicted miREs but did not cause significant repression when corresponding mimics were transfected into the cells, possibly because they are naturally present at a high cellular concentration in the cells. This could lead to an underestimation of the actual number of miREs that are located in the PTC-STOP region. If so, these false-negatives could potentially be identified by repeating the screening in multiple cell lines that have different miRNA profiles, a point that is now made in the Discussion section.

*2) These experiments while elegant need further data on direct effects of miR-29 as there is a possibility here of indirect effects. I think evidence that miR-29 really associates with the APC PTC mutant RNA in vivo needs to be obtained by Ago pull down of the specific mRNA. Alternatively an Ago CLIP analysis might be useful*.

We have performed several experiments to corroborate that miR-29a binds directly to PTC-*APC* mRNA. First, we performed Ago2-RIP assays to determine the change in the abundance of endogenous PTC-*APC* mRNA associated with Ago2 in SW480 cells upon miR-29a knockdown or overexpression. The level of endogenous PTC-*APC* mRNA associated with Ago2 showed a mild but reproducible decrease when miR-29a was knocked down (Figure 3) and a significant increase upon miR-29a overexpression in SW480 cells (Figure 3). The 3’ UTR of *APC* mRNA contains no miR-29a miREs, which was determined by comparing the expression of a luciferase reporter bearing a full length *APC* 3’ UTR in the presence of miR-29a versus a negative control small RNA (siNC) (Figure 3). Therefore, the changes in the Ago2-associated PTC-*APC* mRNA levels in SW480 cells are most likely due to a direct effect of miR-29a targeting its miREs within the PTC-STOP region. Moreover, we used the Ago2-RIP assay to compare the binding efficiency of *APC*-PTC1450 mRNAs bearing wild-type or mutant miREs by co-transfecting HEK293 cells with the minigene constructs and a miR-29a mimic or a control small RNA. We observed a 3-fold enrichment of the PTC-*APC* mRNA bearing wild-type miREs versus mutant miREs, and this enrichment was miR-29a-specific (Figure 3). In addition, we synthesized a miR-29a mutant that restores complementary to the mutant miREs and found that this mutant miRNA successfully restores repression of the PTC-*APC* mRNA bearing mutant miREs (Figure 3—figure supplement 5). These experiments confirm that miR-29a specifically recognizes and binds to the miR-29a miREs in *APC* mRNA at the predicted sites.

Figure 4*:*

*The authors do not consider that in the context of an endogenous PTC-containing mRNA, it is likely that the 3' UTR will be targeted by miRNAs. What would the effects of this targeting be, and would the unmasking of additional sites upstream of the original stop codon make a significant addition to existing 3' UTR-mediated miRNA repression? The constructs used to assess APC and BRCA1 NMD lack the endogenous 3' UTR; it would be important to include this in at least some of the experiments to assess how this influences miRNA-mediated NMD*.

We agree with the reviewers’ concerns and have added several experiments to address this issue, as now shown in Figure 3—figure supplement 2 and Figure 3—figure supplement 3. First, we constructed a pair of *APC*-PTC1450 minigene plasmids that contain the full length 3’ UTR of *APC* mRNA, one with wild-type miR-29a miREs and the other with the mutant miREs. Western blotting showed that, in the context of the natural *APC* 3’ UTR, the PTC was still able to potentiate the repressive effect of miR-29a miREs originally located in the ORF (Figure 3—figure supplement 2). In addition, to examine the relationship between unmasked ORF miREs and existing 3’ UTR miREs more quantitatively, we designed chimeric reporters in which the PTC-STOP region and the entire 3’ UTR of *APC* mRNA were fused to the 3’ end of a firefly luciferase ORF. The miR-29a miREs in the PTC-STOP region and a miR-135b miRE in the *APC* 3’ UTR previously reported to be functional (Nagel et al., *Cancer Res*, 2008) were mutated either individually or simultaneously. In the presence of both miRNAs, the wild-type chimeric reporter was repressed efficiently, while mutating either the ORF miRE or the 3’ UTR miRE alleviated the repression (Figure 3—figure supplement 3). These observations support the conclusion that ORF miREs unmasked by an upstream PTC are fully functional even when miREs in the 3’ UTR are also targeted by miRNAs.